# Path Regularization: A Convexity and Sparsity Inducing Regularization for Parallel ReLU Networks

## Abstract

Understanding the fundamental principles behind the success of deep neural networks is one of the most important open questions in the current literature. To this end, we study the training problem of deep neural networks and introduce an analytic approach to unveil hidden convexity in the optimization landscape. We consider a deep parallel ReLU network architecture, which also includes standard deep networks and ResNets as its special cases. We then show that pathwise regularized training problems can be represented as an exact convex optimization problem. We further prove that the equivalent convex problem is regularized via a group sparsity inducing norm. Thus, a path regularized parallel ReLU network can be viewed as a parsimonious convex model in high dimensions. More importantly, since the original training problem may not be trainable in polynomial-time, we propose an approximate algorithm with a fully polynomial-time complexity in all data dimensions. Then, we prove strong global optimality guarantees for this algorithm. We also provide experiments corroborating our theory.

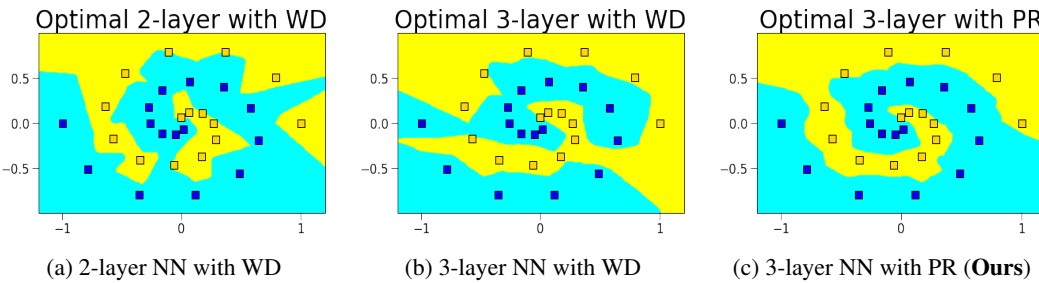

| (a) 2-layer NN with WD | (b) 3-layer NN with WD | (c) 3-layer NN with PR (**Ours**) |
|---|---|---|

Figure 1: Decision boundaries of 2-layer and 3-layer ReLU networks that are globally optimized with weight decay (WD) and path regularization (PR). Here, our convex training approach in (c) successfully learns the underlying spiral pattern for each class while the previously studied convex models in (a) and (b) fail (see Appendix A.1 for details).

## 1 Introduction

Deep Neural Networks (DNNs) have achieved substantial improvements in several fields of machine learning. However, since DNNs have a highly nonlinear and non-convex structure, the fundamental principles behind their remarkable performance is still an open problem. Therefore, advances in this field largely depend on heuristic approaches. One of the most prominent techniques to boost the generalization performance of DNNs is regularizing layer weights so that the network can fit a function that performs well on unseen test data. Even though weight decay, i.e., penalizing the $\ell_2^2$-norm of the layer weights, is commonly employed as a regularization technique in practice, recently, it has been shown that $\ell_2$-path regularizer (Neyshabur et al., 2015b), i.e., the sum over all paths in the network of the squared product over all weights in the path, achieves further empirical gains (Neyshabur et al., 2015a). Therefore, in this paper, we investigate the underlying mechanisms behind path regularized DNNs through the lens of convex optimization.

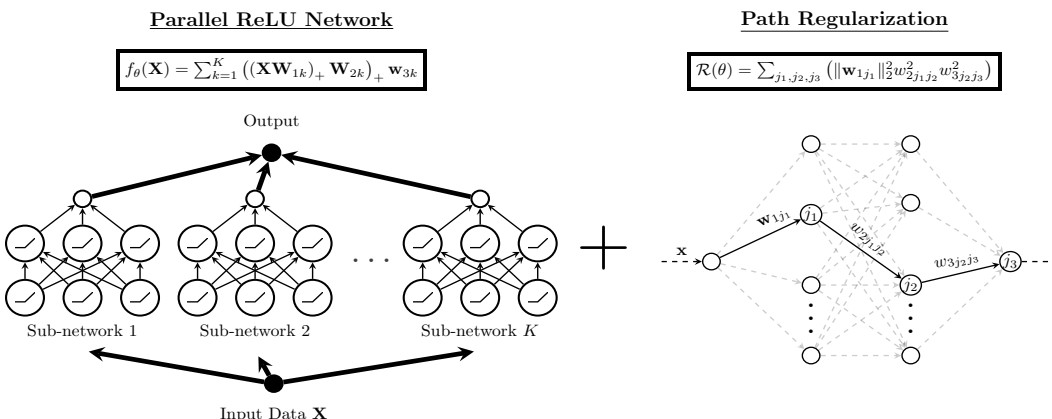

Figure 2: **(Left):** Parallel ReLU network in (1) with $K$ sub-networks and three layers ($L = 3$) **(Right):** Path regularization for a three-layer network.

Table 1: Complexity comparison with prior works ($n$: # of data samples, $d$: feature dimension, $m_l$: # of hidden neurons in layer $l$, $\epsilon$: approximation accuracy, $r$: rank of the data, $\kappa \ll r$: chosen according to (10) such that $\epsilon$ accuracy is achieved)

| | Loss function | 2-layer complexity | $L$-layer complexity | Globally optimal |
|---|---|---|---|---|
| Arora et al. (2018a) | Convex loss | $\mathcal{O}\left(2^{m_1}n^{dm_1}poly(n,d,m_1)\right)$ | - | ✓(Brute-force) |
| Goel et al. (2021) | $\ell_2$-loss | $2^{\mathcal{O}(\frac{m_1^5}{\epsilon^2})}poly(n,d)$ | | ✗(NP-hard) |
| Froese et al. (2021) | $\ell_p$-loss | $\mathcal{O}\left(2^{m_1}n^{dm_1}poly(n,d,m_1)\right)$ | - | ✓(Brute-force) |
| Pilanci & Ergen (2020) | Convex loss | $\mathcal{O}(n^r poly(d,r))$ | - | ✓(Convex, exact) |
| **Ours** | Convex loss | $\mathcal{O}(n^r poly(d,r))$ | $\mathcal{O}\left(n^{r\prod_{j=1}^{L-2}m_j}poly(d,r,\prod_{j=1}^{L-2}m_l)\right)$ | ✓(Convex, exact) |
| **Ours** | Convex loss | $\mathcal{O}(n^\kappa poly(d,\kappa))$ | $\mathcal{O}\left(n^{\kappa\prod_{j=1}^{L-2}m_j}poly(d,\kappa,\prod_{j=1}^{L-2}m_l)\right)$ | ✓(Convex, $\epsilon$-opt) |

## 2 PARALLEL NEURAL NETWORKS

Although DNNs are highly complex architectures due to the composition of multiple nonlinear functions, their parameters are often trained via simple first order gradient based algorithms, e.g., Gradient Descent (GD) and variants. However, since such algorithms only rely on local gradient of the objective function, they may fail to globally optimize the objective in certain cases (Shalev-Shwartz et al., 2017; Goodfellow et al., 2016). Similarly, Ge et al. (2017); Safran & Shamir (2018) showed that these pathological cases also apply to stochastic algorithms such as Stochastic GD (SGD). They further show that some of these issues can be avoided by increasing the number of trainable parameters, i.e., operating in an overparameterized regime. However, Anandkumar & Ge (2016) reported the existence of more complicated cases, where SGD/GD usually fails. Therefore, training DNNs to global optimality remains a challenging optimization problem (DasGupta et al., 1995; Blum & Rivest, 1989; Bartlett & Ben-David, 1999).

To circumvent difficulties in training, recent studies focused on models that benefit from overparameterization (Brutzkus et al., 2017; Du & Lee, 2018; Arora et al., 2018b; Neyshabur et al., 2018). As an example, Wang et al. (2021); Ergen & Pilanci (2021d;c); Zhang et al. (2019); Haeffele & Vidal (2017) considered a new architecture by combining multiple standard NNs, termed as sub-networks, in parallel. Evidences in Ergen & Pilanci (2021d;c); Zhang et al. (2019); Haeffele & Vidal (2017); Zagoruyko & Komodakis (2016); Veit et al. (2016) showed that this way of combining NNs yields an optimization landscape that has fewer local minima and/or saddle points so that SGD/GD generally converges to a global minimum. Therefore, many recently proposed NN-based architectures that achieve state-of-the-art performance in practice, e.g., SqueezeNet (Iandola et al., 2016), Inception (Szegedy et al., 2017), Xception (Chollet, 2017), and ResNext (Xie et al., 2017), are in this form.

**Notation and preliminaries:** Throughout the paper, we denote matrices and vectors as uppercase and lowercase bold letters, respectively. For vectors and matrices, we use subscripts to denote a certain column/element. As an example, $w_{lkj_{l-1}j_l}$ denotes the $j_{l-1}j_l^{th}$ entry of the matrix $\mathbf{W}_{lk}$. We use $\mathbf{I}_k$ and $\mathbf{0}$ (or $\mathbf{1}$) to denote the identity matrix of size $k \times k$ and a vector/matrix of zeros (or ones)

with appropriate sizes. We use $[n]$ for the set of integers from $1$ to $n$. We use $\| \cdot \|_2$ and $\| \cdot \|_F$ to represent the Euclidean and Frobenius norms, respectively. Additionally, we denote the unit $\ell_p$ ball as $\mathcal{B}_p := \{ \mathbf{u} \in \mathbb{R}^d : \|\mathbf{u}\|_p \le 1 \}$. We also use $\mathbb{1}[x \ge 0]$ and $(x)_+ = \max\{x, 0\}$ to denote the 0-1 valued indicator and ReLU, respectively.

In this paper, we particularly consider a parallel ReLU network with $K$ sub-networks and each sub-network is an $L$-layer ReLU network (see Figure 2) with layer weights $\mathbf{W}_{lk} \in \mathbb{R}^{m_{l-1} \times m_l}$, $\forall l \in [L-1]$ and $\mathbf{w}_{Lk} \in \mathbb{R}^{m_{L-1}}$, where $m_0 = d, m_L = 1$[1], and $m_l$ denotes the number of neurons in the $l^{th}$ hidden layer. Then, given a data matrix $\mathbf{X} \in \mathbb{R}^{n \times d}$, the output of the network is as follows

$$f_\theta(\mathbf{X}) := \sum_{k=1}^{K} \left( (\mathbf{X}\mathbf{W}_{1k})_+ \ldots \mathbf{W}_{(L-1)k} \right)_+ \mathbf{w}_{Lk}, \tag{1}$$

where we compactly denote the parameters as $\theta := \bigcup_k \{\mathbf{W}_{lk}\}_{l=1}^{L}$ with the parameter space as $\Theta$ and each sub-network represents a standard deep ReLU network.

**Remark 1.** *Most commonly used neural networks in practice can be classified as special cases of parallel networks, e.g., standard NNs and ResNets (He et al., 2016) see Appendix A.2 for details.*

## 2.1 OUR CONTRIBUTIONS

- We prove that training the path regularized parallel ReLU networks (1) is equivalent to a convex optimization problem that can be approximately solved in polynomial-time by standard convex solvers (see Table 1). Therefore, we generalize the two-layer results in Pilanci & Ergen (2020) to multiple nonlinear layer without any strong assumptions in contrast to Ergen & Pilanci (2021c) and a much broader class of NN architectures including ResNets.

- As already observed by Pilanci & Ergen (2020); Ergen & Pilanci (2021c), regularized deep ReLU network training problems require exponential-time complexity when the data matrix is full rank, which is unavoidable. However, in this paper, we develop an approximate training algorithm which has fully polynomial-time complexity in all data dimensions and prove global optimality guarantees in Theorem 2. To the best of our knowledge, this is the first convex optimization based and polynomial-time complexity (in data dimensions) training algorithm for ReLU networks with global approximation guarantees.

- We show that the equivalent convex problem is regularized by a group norm regularization where grouping effect is among the sub-networks. Therefore the equivalent convex formulation reveals an implicit regularization that promotes group sparsity among sub-networks and generalizes prior works on linear networks such as Dai et al. (2021) to ReLU networks.

- We derive a closed-form mapping between the parameters of the non-convex parallel ReLU networks and its convex equivalent in Proposition 1. Therefore, instead of solving the challenging non-convex problem, one can globally solve the equivalent convex problem and then construct an optimal solution to the original non-convex network architecture via our closed-form mapping.

## 2.2 OVERVIEW OF OUR RESULTS

Given data $\mathbf{X} \in \mathbb{R}^{n \times d}$ and labels $\mathbf{y} \in \mathbb{R}^n$, we consider the following regularized training problem

$$p_L^* := \min_{\theta \in \Theta} \mathcal{L}\left(f_\theta(\mathbf{X}), \mathbf{y}\right) + \beta \mathcal{R}(\theta), \tag{2}$$

where $\Theta := \{\theta \in \Theta : \mathbf{W}_{lk} \in \mathbb{R}^{m_{l-1} \times m_l}, \forall l \in [L], \forall k \in [K]\}$ is the parameter space, $\mathcal{L}(\cdot, \cdot)$ is an arbitrary convex loss function, $\mathcal{R}(\cdot)$ represents the regularization on the network weights, and $\beta > 0$ is the regularization coefficient.

For the rest of the paper, we focus on a scalar output regression/classification framework with arbitrary loss functions, e.g., squared loss, cross entropy or hinge loss. We also note that our derivations can be straightforwardly extended to vector outputs networks as proven in Appendix A.11. More importantly, we use $\ell_2$-path regularizer studied in Neyshabur et al. (2015b;a), which is defined as

$$\mathcal{R}(\theta) := \sum_{k=1}^{K} \sqrt{\sum_{j_1, j_2, \ldots, j_L} \left( \|\mathbf{w}_{1kj_1}\|_2^2 \prod_{l=2}^{L} w_{lkj_{l-1}j_l}^2 \right)},$$

---

[1] We analyze scalar outputs, however, our derivations extend to vector outputs as shown in Appendix A.11.

where $w_{lkj_{l-1}j_l}$ is the $j_{l-1}j_l^{th}$ entry of $\mathbf{W}_{lk}$. The above regularizer sums the square of all the parameters along each possible path from input to output of each sub-network $k$ (see Figure 2) and then take the squared root of the summation. Therefore, we penalize each path in each sub-network and then group them based on the sub-network index $k$.

We now propose a scaling to show that (2) is equivalent to a group $\ell_1$ regularized problem.

**Lemma 1.** *The following problems are equivalent* [2]*:*

$$\min_{\theta \in \Theta} \mathcal{L}(f_\theta(\mathbf{X}), \mathbf{y}) + \beta \mathcal{R}(\theta) = \min_{\theta \in \Theta_s} \mathcal{L}(f_\theta(\mathbf{X}), \mathbf{y}) + \beta \sum_{k=1}^{K} \|\mathbf{w}_{Lk}\|_2,$$

*where $\mathbf{w}_{Lk} \in \mathbb{R}^{m_{L-1}}$ are the last layer weights of each sub-network $k$, and $\Theta_s := \{\theta \in \Theta : \sum_{j_1, j_2, \ldots, j_{L-2}} \left( \|\mathbf{w}_{1kj_1}\|_2^2 \prod_{l=2}^{L-1} w_{lkj_{l-1}j_l}^2 \right) \leq 1, \forall j_{L-1}, \forall k \in [K]\}$ denotes the parameter space after rescaling.*

The advantage of the form in Lemma 1 is that we can derive a dual problem with respect to the output layer weights $\mathbf{w}_{Lk}$ and then characterize the optimal layer weights via optimality conditions and the prior works on $\ell_1$ regularization in infinite dimensional spaces (Rosset et al., 2007). Thus, we first apply the rescaling in Lemma 1 and then take the dual with respect to the output weights $\mathbf{w}_{Lk}$. To characterize the hidden layer weights, we then change the order of minimization for the hidden layer weights and the maximization for the dual parameter to get the following dual problem[3]

$$p_L^* \geq d_L^* := \max_{\mathbf{v}} -\mathcal{L}^*(\mathbf{v}) \quad \text{s.t.} \max_{\theta \in \Theta_s} \left\| \mathbf{v}^T \left( (\mathbf{X}\mathbf{W}_1)_+ \ldots \mathbf{W}_{(L-1)} \right)_+ \right\|_2 \leq \beta, \tag{3}$$

where $\mathcal{L}^*$ is the Fenchel conjugate function of $\mathcal{L}$, which is defined as (Boyd & Vandenberghe, 2004)

$$\mathcal{L}^*(\mathbf{v}) := \max_{\mathbf{z}} \mathbf{z}^T \mathbf{v} - \mathcal{L}(\mathbf{z}, \mathbf{y}).$$

The dual problem in (3) is critical for our derivations since it provides us with an analytic perspective to characterize a set of optimal hidden layer weights for the non-convex neural network in (1). To do so, we first show that strong duality holds for the non-convex training problem in Lemma 1, i.e., $p_L^* = d_L^*$. Then, based on the exact dual problem in (3), we propose an equivalent analytic description for the optimal hidden layer weights via the KKT conditions.

We note that strong duality for two-layer ReLU networks has already been proved by previous studies (Wang et al., 2021; Ergen & Pilanci, 2021c; Pilanci & Ergen, 2020; Ergen & Pilanci, 2021b; Zhang et al., 2019; Bach, 2017), however, this is the first work providing an exact characterization for path regularized deep ReLU networks via convex duality.

## 3 PARALLEL NETWORKS WITH THREE LAYERS

Here, we consider a three-layer parallel network with $K$ sub-networks, which is a special case of (1) when $L = 3$. Thus, we have the following training problem

$$p_3^* = \min_{\theta \in \Theta} \mathcal{L}(f_\theta(\mathbf{X}), \mathbf{y}) + \beta \sum_{k=1}^{K} \sqrt{\sum_{j_1, j_2} \|\mathbf{w}_{1kj_1}\|_2^2 w_{2kj_1j_2}^2 w_{3kj_2}^2}. \tag{4}$$

where $\Theta = \{(\mathbf{W}_{1k}, \mathbf{W}_{2k}, \mathbf{w}_{3k}) : \mathbf{W}_{1k} \in \mathbb{R}^{d \times m_1}, \mathbf{W}_{2k} \in \mathbb{R}^{m_1 \times m_2}, \mathbf{w}_{3k} \in \mathbb{R}^{m_2}\}$. By Lemma 1,

$$p_3^* = \min_{\theta \in \Theta_s} \mathcal{L}(f_\theta(\mathbf{X}), \mathbf{y}) + \beta \sum_{k=1}^{K} \|\mathbf{w}_{3k}\|_2. \tag{5}$$

Then, taking the dual of (5) with respect to the output layer weights $\mathbf{w}_{3k} \in \mathbb{R}^{m_2}$ and then changing the order of the minimization for $\{\mathbf{W}_{1k}, \mathbf{W}_{2k}\}$ and the maximization for the dual variable $\mathbf{v}$ yields

$$p_3^* \geq d_3^* := \max_{\mathbf{v}} -\mathcal{L}^*(\mathbf{v}) \quad \text{s.t.} \max_{\theta \in \Theta_s} \left\| \mathbf{v}^T \left( (\mathbf{X}\mathbf{W}_1)_+ \mathbf{W}_2 \right)_+ \right\|_2 \leq \beta. \tag{6}$$

---

[2] All the proofs are presented in the supplementary file.

[3] We present the details in Appendix A.7.

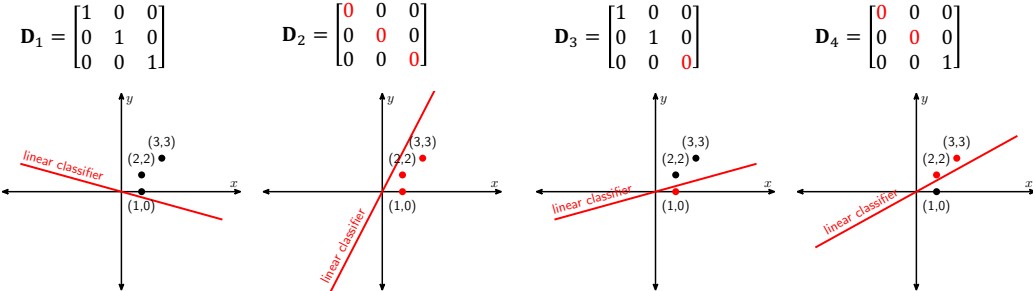

Figure 3: Illustration of possible hyperplane arrangements that determine the diagonal matrices $\mathbf{D}_i$. Here, we have three samples in two dimensions and we want to separate these samples with a linear classifier. $\mathbf{D}_i$ basically encodes information regarding which side of the linear classifier samples lie.

Here, we remark that (4) is non-convex with respect to the layer weights, we may have a duality gap, i.e., $p_3^* \geq d_3^*$. Therefore, we first show that strong duality holds in this case, i.e., $p_3^* = d_3^*$ as detailed in Appendix A.4. We then introduce an equivalent representation for the ReLU activation as follows.

Since ReLU masks the negative entries of inputs, we have the following equivalent representation

$$\left((\mathbf{X}\mathbf{W}_1)_+ \mathbf{w}_2\right)_+ = \left(\sum_{j_1=1}^{m_1} (\mathbf{X}\mathbf{w}_{1j_1})_+ w_{2j_1}\right)_+ = \left(\sum_{j_1=1}^{m_1} (\mathbf{X}\bar{\mathbf{w}}_{j_1})_+ \mathcal{I}_{j_1}\right)_+ = \mathbf{D}_2 \sum_{j_1=1}^{m_1} \mathcal{I}_{j_1} \mathbf{D}_{1j_1} \mathbf{X}\bar{\mathbf{w}}_{j_1},$$
(7)

where $\bar{\mathbf{w}}_{j_1} = |w_{2j_1}|\mathbf{w}_{1j_1}$, $\mathcal{I}_{j_1} = \mathrm{sign}(w_{2j_1}) \in \{-1, +1\}$, and we use the following alternative representation for ReLU (see Figure 3 for a two dimensional visualization)

$$(\mathbf{X}\mathbf{w})_+ = \mathbf{D}\mathbf{X}\mathbf{w} \iff \begin{matrix} \mathbf{D}\mathbf{X}\mathbf{w} \geq 0 \\ (\mathbf{I}_n - \mathbf{D})\mathbf{X}\mathbf{w} \leq 0 \end{matrix} \iff (2\mathbf{D} - \mathbf{I}_n)\mathbf{X}\mathbf{w} \geq 0,$$

where $\mathbf{D} \in \mathbb{R}^{n \times n}$ is a diagonal matrix of zeros/ones, i.e., $\mathbf{D}_{ii} \in \{0, 1\}$. Therefore, we first enumerate all possible signs and diagonal matrices for the ReLU layers and denote them as $\mathcal{I}_{j_1}$, $\mathbf{D}_{1ij_1}$ and $\mathbf{D}_{2l}$ respectively, where $j_1 \in [m_1], i \in [P_1], l \in [P_2]$. Here, $\mathbf{D}_{1ij_1}$ and $\mathbf{D}_{2l}$ denotes the masking/diagonal matrices for the first and second ReLU layers, respectively and $P_1$ and $P_2$ are the corresponding total number diagonal matrices in each layer as detailed in Section A.10. Then, we convert the non-convex dual constraints in (6) to a convex constraint using $\mathbf{D}_{1ij_1}$, $\mathbf{D}_{2l}$ and $\mathcal{I}_{j_1}$.

Using the representation in (7), we then take the dual of (6) to obtain the convex bidual of the primal problem (4) as detailed in the theorem below.

**Theorem 1.** *The non-convex training problem in* (4) *can be cast as the following convex program*

$$\min_{\mathbf{z}, \mathbf{z}' \in \mathcal{C}} \mathcal{L}\left(\tilde{\mathbf{X}}(\mathbf{z} - \mathbf{z}'), \mathbf{y}\right) + \frac{\beta}{\sqrt{m_2}}(\|\mathbf{z}\|_{F,1} + \|\mathbf{z}'\|_{F,1}),$$
(8)

*where $\|\cdot\|_{F,1}$ denotes a $d \times m_1$ dimensional group Frobenius norm operator such that given a vector $\mathbf{u} \in \mathbb{R}^{dm_1 P}$, $\|\mathbf{u}\|_{F,1} := \sum_{i=1}^{P} \|\mathbf{U}_i\|_F$, where $\mathbf{U}_i \in \mathbb{R}^{d \times m_1}$ are reshaped partitions of $\mathbf{u}$. Moreover, the convex set $\mathcal{C}$ is defined as*

$$\mathcal{C} := \{\mathbf{z} : \mathbf{z}_{ij_1 l}^s \in \mathcal{C}_{il}^s, \forall i \in [P_1], l \in [P_2], s \in [M]\}$$

$$\mathcal{C}_{il}^s := \left\{\{\mathbf{w}_{j_1}\}_{j_1} : (2\mathbf{D}_{2l} - \mathbf{I}_n)\sum_{j_1=1}^{m_1} \mathcal{I}_{ij_1 l}^s \mathbf{D}_{1ij_1} \mathbf{X}\mathbf{w}_{j_1} \geq 0, (2\mathbf{D}_{1ij_1} - \mathbf{I}_n)\mathbf{X}\mathbf{w}_{j_1} \geq 0, \forall j_1 \in [m_1]\right\}$$

*where $\mathcal{I}_{ij_1 l}^s \in \{+1, -1\}$, $M = 2^{m_1}$, and $\mathbf{z}, \mathbf{z}' \in \mathbb{R}^{dm_1 M P_1 P_2}$ are constructed by stacking $\mathbf{z}_{ij_1 l}^s, \mathbf{z}_{ij_1 l}^{s'} \in \mathbb{R}^d$, $\forall i \in [P_1], l \in [P_2], j_1 \in [m_1], s \in [M]$, respectively. Also, the effective data matrix $\tilde{\mathbf{X}} \in \mathbb{R}^{n \times dm_1 M P_1 P_2}$ is defined as*

$$\tilde{\mathbf{X}} := \mathbf{I}_M \otimes \tilde{\mathbf{X}}_s, \qquad \tilde{\mathbf{X}}_s := [\mathbf{D}_{21}\mathbf{D}_{111}\mathbf{X} \dots \mathbf{D}_{2l}\mathbf{D}_{1ij_1}\mathbf{X} \dots \mathbf{D}_{2P_2}\mathbf{D}_{1P_1 m_1}\mathbf{X}].$$

We next derive a mapping between the convex program (8) and the non-convex architecture (4).

**Proposition 1.** *An optimal solution to the non-convex parallel network training problem in (4), denoted as $\{\mathbf{W}_{1k}^*, \mathbf{w}_{2k}^*, w_{3k}^*\}_{k=1}^K$, can be recovered from an optimal solution to the convex program in (8), i.e., $\{\mathbf{z}^*, \mathbf{z'}^*\}$ via a closed-form mapping. Therefore, we prove a mapping between the parameters of the parallel network in Figure 2 and its convex equivalent.*

Next, we prove that the convex program in (8) can be globally optimized with a polynomial-time complexity given $\mathbf{X}$ has fixed rank, i.e., $\text{rank}(\mathbf{X}) = r < \min\{n, d\}$.

**Proposition 2.** *Given a data matrix such that $\text{rank}(\mathbf{X}) = r < \min\{n, d\}$, the convex program in (8) can be globally optimized via standard convex solvers with $\mathcal{O}(d^3 m_1^3 m_2^3 2^{3(m_1+1)m_2} n^{3(m_1+1)r})$ complexity, which is a polynomial-time complexity in terms of $n, d$. Note that here globally optimizing the training objective means to achieve the exact global minimum up to any arbitrary machine precision or solver tolerance.*

Below, we show that the complexity analysis in Proposition 2 extends to arbitrarily deep networks.

**Corollary 1.** *The same analysis can be readily applied to arbitrarily deep networks. Therefore, given $\text{rank}(\mathbf{X}) = r < \min\{n, d\}$, we prove that $L$-layer architectures can be globally optimized with $\mathcal{O}\left(d^3 \left(\prod_{j=1}^{L-2} m_j^3\right) 2^{3\sum_{j=1}^{L-1} m_j} n^{3r\left(1+\sum_{l=1}^{L-2} \prod_{j=1}^{l} m_j\right)}\right)$, which is polynomial in $n, d$.*

### 3.1 POLYNOMIAL-TIME TRAINING FOR ARBITRARY DATA

Based on the analysis in Corollary 1, exponential complexity is unavoidable for deep networks when the data matrix is full rank, i.e., $\text{rank}(\mathbf{X}) = \min\{n, d\}$. Thus, we propose a low rank approximation to the model in (4). We first denote the rank-$r$ approximation of $\mathbf{X}$ as $\hat{\mathbf{X}}_r$ such that $\|\mathbf{X} - \hat{\mathbf{X}}_r\|_2 \leq \sigma_{r+1}$, where $\sigma_{r+1}$ represents the $(r+1)^{th}$ largest singular value of $\mathbf{X}$. Then, we have the following result.

**Theorem 2.** *Given an $R$-Lipschitz convex loss function $\mathcal{L}(\cdot, \mathbf{y})$, the regularized training problem*

$$p_3^* = \min_{\theta \in \Theta} \mathcal{L}(f_\theta(\mathbf{X}), \mathbf{y}) + \beta \sum_{k=1}^K \sqrt{\sum_{j_1, j_2} \|\mathbf{w}_{1kj}\|_2^2 w_{2kj_1 j_2}^2 w_{3kj_2}^2}, \tag{9}$$

*can be solved using the data matrix $\hat{\mathbf{X}}_r$ to achieve the following optimality guarantee*

$$p_3^* \leq p_r \leq p_3^* \left(1 + \frac{\sqrt{m_1 m_2} R \sigma_{r+1}}{\beta}\right)^2, \tag{10}$$

*where $p_r$ denotes the objective value achieved by the parameters trained using $\hat{\mathbf{X}}_r$.*

**Remark 2.** *Theorem 1 and 2 imply that for a given arbitrary rank data matrix $\mathbf{X}$, the regularized training problem in (4) can be approximately solved by convex solvers to achieve a worst-case approximation $p_3^* \left(1 + \frac{RR\sigma_{r+1}}{\beta}\right)^2$ with complexity $\mathcal{O}(d^3 m_1^3 2^{3(m_1+1)} n^{3(m_1+1)r})$, where $r \ll \min\{n, d\}$. Therefore, even for full rank data matrices where the complexity is exponential in $n$ or $d$, one can approximately solve the convex program in (8) in polynomial-time. Moreover, we remark that the approximation error proved in Theorem 2 can be arbitrarily small for practically relevant problems. As an example, consider a parallel network training problem with $\ell_2$ loss function, then the upperbound becomes $(1 + \frac{\sqrt{m_1 m_2} \sigma_{r+1}}{\beta})^2$, which is typically close to one due to fast decaying singular values in practice (see Figure 4).*

### 3.2 REPRESENTATIONAL POWER: TWO VERSUS THREE LAYERS

Here, we provide a complete explanation for the representational power of three-layer networks by comparing with the two-layer results in Pilanci & Ergen (2020). We first note that three-layer networks have substantially higher expressive power due to the non-convex interactions between hidden layers as detailed in Allen-Zhu et al. (2019); Pham & Nguyen (2021). Furthermore, Belilovsky et al. (2019) show that layerwise training of three-layer networks can achieve comparable performance

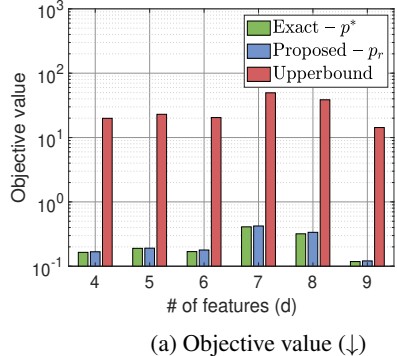
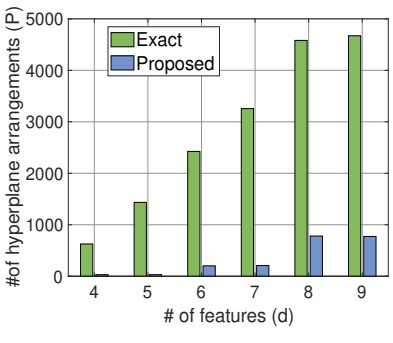

(a) Objective value ($\downarrow$)          (b) # of hyperplane arrangements ($P$)

Figure 4: Verification of Theorem 2 and Remark 2. We train a parallel network using the convex program in Theorem 1 with $\ell_2$ loss on a toy dataset with $n = 15$, $\beta = 0.1$, $m_2 = 1$, and the low-rank approximation $r = \lfloor \frac{d}{2} \rfloor$. To obtain a low-rank model, we first sample a data matrix from a standard normal Gaussian distribution and then set $\sigma_{r+1} = \ldots = \sigma_d = 1$.

Table 2: Training objective of a three-layer parallel network trained with non-convex SGD (5 independent initialization trials) on a toy dataset with $(n, d, m_1, m_2, \beta, \text{batch size}) = (5, 2, 3, 1, 0.002, 5)$, where the convex program in (8) are solved via the interior point solvers in CVX/CVXPY.

| Method | | Non-convex SGD | | | | | Convex(Ours) |
|---|---|---|---|---|---|---|---|
| | | Run #1 | Run #2 | Run #3 | Run #4 | Run #5 | |
| $K = 5$ | Training objective | 0.0221 | 0.0239 | 0.0031 | 0.0027 | 0.0027 | **0.0007** |
| | Time(s) | 11.62 | 11.62 | 11.62 | 11.62 | 11.62 | **4.947** |
| $K = 20$ | Training objective | 0.0010 | 0.0027 | 0.0009 | 0.0010 | 0.0027 | **0.0007** |
| | Time(s) | 44.37 | 44.37 | 44.37 | 11.55 | 44.37 | **4.947** |
| $K = 40$ | Training objective | 0.0008 | 0.0009 | 0.0009 | 0.0009 | 0.0008 | **0.0007** |
| | Time(s) | 91.87 | 91.87 | 91.87 | 91.87 | 91.87 | **4.947** |

to deeper models, e.g., VGG-11, on Imagenet. There exist several studies analyzing two-layer networks, however, despite their empirical success, a full theoretical understanding and interpretation of three-layer networks is still lacking in the literature. In this work, we provide a complete characterization for three-layer networks through the lens of convex optimization theory. To understand their expressive power, we compare our convex program for three-layer networks in (8) with its two-layer counterpart in Pilanci & Ergen (2020).

Pilanci & Ergen (2020) analyzes two-layer networks with one ReLU layer, so that the data matrix $\mathbf{X}$ is multiplied with a single diagonal matrix (or hyperplane arrangement) $\mathbf{D}_i$. Thus, the effective data matrix is in the form of $\tilde{\mathbf{X}}_s = [\mathbf{D}_1 \mathbf{X} \ldots \mathbf{D}_P \mathbf{X}]$. However, since our convex program in (8) has two nonlinear ReLU layers, the composition of these two-layer can generate substantially more complex features via locally linear variables $\{\mathbf{w}^s_{ij_1l}\}$ multiplying the $d$-dimensional blocks of the columns of the effective data matrix $\tilde{\mathbf{X}}_s$ in Theorem 1. Although this may seem similar to the features in Ergen & Pilanci (2021c), here, we have $2^{m_1}$ variables for each linear region unlike Ergen & Pilanci (2021c) which employ 2 variables per linear region. Moreover, Ergen & Pilanci (2021c) only considers the case where the second hidden layer has only one neuron, i.e., $m_2 = 1$, therefore do not consider standard three layer or deeper networks. Hence, we exactly describe the impact of having one more ReLU layer and its contribution to the representational power of the network.

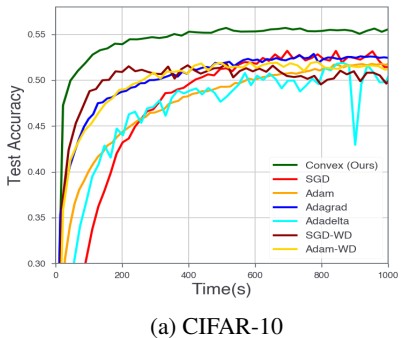

(a) CIFAR-10

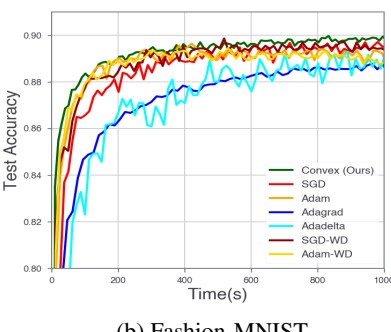

(b) Fashion-MNIST

Figure 5: Accuracy of a three-layer architecture trained using the non-convex formulation (4) and the proposed convex program (8), where we use **(a)** CIFAR-10 with $(n, d, m_1, m_2, K, \beta, \text{batch size}) = (5\text{x}10^4, 3072, 100, 1, 40, 10^{-3}, 10^3)$ and **(b)** Fashion-MNIST with $(n, d, m_1, m_2, K, \beta, \text{batch size}) = (6\text{x}10^4, 784, 100, 1, 40, 10^{-3}, 10^3)$. We note that the convex model is trained using **(a)** SGD and **(b)** Adam.

## 4 EXPERIMENTS

In this section[4], we present numerical experiments corroborating our theory.

**Low rank model in Theorem 2:** To validate our claims, we generate a synthetic dataset as follows. We first randomly generate a set of layer weights for a parallel ReLU network with $K = 5$ sub-networks by sampling from a standard normal distribution. We then obtain the labels as $\mathbf{y} = \sum_k \left( (\mathbf{XW}_{1k})_+ \mathbf{W}_{2k} \right)_+ \mathbf{w}_{3k} + 0.1\boldsymbol{\epsilon}$, where $\boldsymbol{\epsilon} \sim N(\mathbf{0}, \mathbf{I}_n)$. To promote a low rank structure in the data, we first sample a matrix from the standard normal distribution and then set $\sigma_{r+1} = \ldots = \sigma_d = 1$. We consider a regression framework with $\ell_2$ loss and $(n, r, \beta, m_1, m_2) = (15, d/2, 0.1, 1, 1)$ and present the numerical results in Figure 4. Here, we observe that the low rank approximation of the objective $p_r$ is closer to $p_3^*$ than the worst-case upper-bound predicted by Theorem 2. However, in Figure 4b, the low rank approximation provides a significant reduction in the number of hyperplane arrangements, and therefore in the complexity of solving the convex program.

**Toy dataset:** We use a toy dataset with 5 samples and 2 features, i.e., $(n, d) = (5, 2)$. To generate the dataset, we forward propagate i.i.d. samples from a standard normal distribution, i.e., $\mathbf{x}_i \sim \mathcal{N}(\mathbf{0}, \mathbf{I}_d)$, through a parallel network with 3 layers, 5 sub-networks, and 3 neurons, i.e., $(L, K, m_1, m_2) = (3, 5, 3, 1)$. We then train the parallel network in (4) on this toy dataset using both our convex program (8) and non-convex SGD. We provide the training objective and wall-clock time in Table 2, where we particularly include 5 initialization trials for SGD. This experiment shows that when the number of sub-networks $K$ is small, SGD trials fail to converge to the global minimum achieved by our convex program. However, as we increase $K$, the number of trials converging to global minimum gradually increases. Therefore, we show the benign overparameterization impact.

**Image classification:** We conduct experiments on benchmark image datasets, namely CIFAR-10 (Krizhevsky et al., 2014) and Fashion-MNIST (Xiao et al., 2017). We particularly consider a ten class classification task and use a parallel network with 40 sub-networks and 100 hidden neurons, i.e., $(K, m_1, m_2) = (40, 100, 1)$. In Figure 5, we plot the test accuracies against wall-clock time, where we include several different optimizers as well as SGD. Moreover, we include a parallel network trained with SGD/Adam and Weight Decay (WD) regularization to show the effectiveness of path regularization in (4). We first note that our convex approach achieves both faster convergence and higher final test accuracies for both dataset. However, the performance gain for Fashion-MNIST seems to be significantly less compared to the CIFAR-10 experiment. This is due to the nature of these datasets. More specifically, since CIFAR-10 is a much more challenging dataset, the baseline accuracies are quite low (around ∼ 50%) unlike Fashion-MNIST with the baseline accuracies around ∼ 90%. Therefore, the accuracy improvement achieved by the convex program seems low in Figure 5b. We also observe that weight decay achieves faster convergence rates however path

---

[4]Details on the experiments can be found in Appendix A.1.

regularization yields higher final test accuracies. It is normal to have faster convergence with weight decay since it can be incorporated into gradient-based updates without any computational overhead.

## 5 RELATED WORK

Parallel neural networks were previously investigated by Zhang et al. (2019); Haeffele & Vidal (2017). Although these studies provided insights into the solutions, they require assumptions, e.g., sparsity among sub-networks in Theorem 1 of Haeffele & Vidal (2017)) and linear activations and hinge loss assumptions in Zhang et al. (2019), which invalidates applications in practice.

Recently, Pilanci & Ergen (2020) studied weight decay regularized two-layer ReLU network training problems and introduced polynomial-time trainable convex formulations. However, their analysis is restricted to standard two-layer ReLU networks, i.e., in the form of $f_\theta(\mathbf{X}) = (\mathbf{X}\mathbf{W}_1)_+ \mathbf{w}_2$. The reasons for this restriction is that handling more than one ReLU layer is a substantially more challenging optimization problem. As an example, a direct extension of Pilanci & Ergen (2020) to three-layer NNs will yield doubly exponential complexity, i.e., $\mathcal{O}(n^{rn^r})$ for a rank-$r$ data matrix, due to the combinatorial behavior of multiple ReLU layers. Thus, they only examined the case with a single ReLU layer (see Table 1 for details and the other relevant references in Pilanci & Ergen (2020)). In addition, since they only considered standard two-layer ReLU networks, their analysis is not valid for a broader range of NN-based architectures as detailed in Remark 1. Later on, Ergen & Pilanci (2021c) extended this approach to three-layer ReLU networks. However, since they analyzed $\ell_2$-norm regularized training problem, they had to put unit Frobenius norm constraints on the first layer weights, which does not reflect the settings in practice. In addition, their analysis is restricted to the networks with a single neuron in the second layer (i.e. $m_2 = 1$) that is in the form of $f_\theta(\mathbf{X}) = \sum_k \left((\mathbf{X}\mathbf{W}_{1k})_+ \mathbf{w}_{2k}\right)_+ w_{3k}$. Since this architecture only allows a single neuron in the second layer, each sub-network $k$ has an expressive power that is equivalent to a standard two-layer network rather than three-layer. This can also be realized from the definition of the constraint set $\mathcal{C}$ in Theorem 1. Specifically, the convex set $\mathcal{C}$ in Ergen & Pilanci (2021c) has decoupled constraints across the hidden layer index $j_1$ whereas our formulation sums the responses over hidden neurons before feeding through the next layer as standard deep networks do. Therefore, this analysis does not reflect the true power of deep networks with $L > 2$. Moreover, the approach in Ergen & Pilanci (2021c) has exponential complexity when the data matrix has full rank, which is unavoidable.

However, we analyze deep neural networks in (1) without any assumption on the weights. Furthermore, we develop an approximate training algorithm which has fully polynomial-time complexity in data dimensions and prove strong global optimality guarantees for this algorithm in Theorem 2.

## 6 CONCLUDING REMARKS

We studied the training problem of path regularized deep parallel ReLU networks, which includes ResNets and standard deep ReLU networks as its special cases. We first showed that the non-convex training problem can be equivalently cast as a single convex optimization problem. Therefore, we achieved the following advantages over the training on the original non-convex formulation: 1) Since our model is convex, it can be globally optimized via standard convex solvers whereas the non-convex formulation trained with optimizers such as SGD might be stuck at a local minimum, 2) Thanks to convexity, our model does not require any sort of heuristics and additional tricks such as learning rate schedule and initialization scheme selection or dropout. More importantly, we proposed an approximation to the convex program to enable fully polynomial-time training in terms of the number of data samples $n$ and feature dimension $d$. Thus, we proved the polynomial-time trainability of deep ReLU networks without requiring any impractical assumptions unlike Pilanci & Ergen (2020); Ergen & Pilanci (2021c). Notice that we derive an exact convex program only for three-layer networks, however, recently Wang et al. (2021) proved that strong duality holds for arbitrarily deep parallel networks. Therefore, a similar analysis can be extended to deeper networks to achieve an equivalent convex program, which is quite promising for future work. Additionally, although we analyzed fully connected networks in this paper, our approach can be directly extended to various NN architectures, e.g., convolution networks (Ergen & Pilanci, 2021a), generative adversarial networks (Sahiner et al., 2021a), NNs with batch normalization (Ergen et al., 2021), and autoregressive models (Gupta et al., 2021).

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

# Supplementary Material

## Table of Contents

## A  APPENDIX

### A.1  ADDITIONAL NUMERICAL RESULTS AND DETAILS

In this section, we provide new numerical results and detailed information about our experiments in the main paper.

**Decision boundary plots in Figure 1:** In order to visualize the capabilities of our convex training approach, we perform an experiment on the spiral dataset which is known to be challenging for 2-layer networks while 3-layer networks can readily interpolate the training data (i.e. exactly fit the training labels) (Carter & Shan). As the baselines of our analysis, we include the two-layer convex training approach in Pilanci & Ergen (2020) and recently introduced three-layer convex training approach (with weight decay regularization and several architectural and parametric assumptions) in Ergen & Pilanci (2021c). As our experimental setup, we consider a binary classification task with $\mathbf{y} \in \{+1, -1\}^n$ and squared loss. We choose $(n, m_1, m_2, P_1, P_2, \beta) = (30, 5, 1, 11, 11, 1e-4)$ and for Pilanci & Ergen (2020) we use $P = 50$ neurons/hyperplane arrangements. We also use CVPXY with MOSEK solver (Grant & Boyd, 2014; Diamond & Boyd, 2016; Agrawal et al., 2018) to globally solve the convex programs. As demonstrated in Figure 1, baselines methods, especially Ergen & Pilanci (2021c), fit a function that is significantly different than the underlying spiral data distribution. This clearly shows that since Pilanci & Ergen (2020) is restricted two-layer networks and Ergen & Pilanci (2021c) have multiple assumptions, i.e., unit Frobenius norm constraints on the layer weights ($\|\mathbf{W}_{lk}\|_F \leq 1, \forall l \in [L-2]$) and last hidden layer weights cannot be matrices ($\mathbf{w}_{(L-1)k} \in \mathbb{R}^{m_{L-2}}$), both baseline approaches fail to reflect true expressive power of deep networks ($L > 2$). On the other hand, our convex training approach for path regularized networks fits a model that successfully describes the underlying data distribution for this challenging task.

**Additional experiments:** We also conduct experiments on several datasets available in UCI Machine Learning Repository (Dua & Graff, 2017), where we particularly selected the datasets from Fernández-Delgado et al. (2014) such that $n \leq 500$. For these datasets, we consider a conventional binary classification framework with $(m_1, m_2, K, \beta) = (100, 1, 40, 0.5)$ and compare the test accuracies of non-convex architectures trained with SGD and Adam with their convex counter parts in (8). For these experiments, we use the $80\% - 20\%$ splitting ratio for the training and test sets. Furthermore, we train each algorithm long enough to reach training accuracy one. As shown in Table 3, our convex approach achieves higher or the same test accuracy compared to the standard non-convex training approach for most of the datasets (precisely 20 and 19 out of 21 datasets for SGD and Adam, respectively). We also note that for this experiment, we used the unconstrained form in (11) with the approximate version in Remark 3.3 of Pilanci & Ergen (2020).

Table 3: Test accuracies for UCI experiments ($(m_1, m_2, K, \beta) = (100, 1, 40, 0.5)$ and $80\% - 20\%$ training-test split). Here, we present the standard non-convex architectures and the proposed convex counterpart trained with SGD and Adam optimizers. If one approach achieves higher accuracy on a certain dataset, we display the corresponding accuracy value in bold font. We observe that our convex approach achieves either higher or the same accuracy for 20 and 19 datasets (out of 21 datasets) when trained with SGD and Adam, respectively

| | | | SGD | | Adam | |
|---|---|---|---|---|---|---|
| **Dataset** | $n$ | $d$ | **Non-convex** | **Convex(Ours)** | **Non-convex** | **Convex(Ours)** |
| acute-inflammation | 120 | 6 | 1.000 | 1.000 | 1.000 | 1.000 |
| acute-nephritis | 120 | 6 | 1.000 | 1.000 | 1.000 | 1.000 |
| balloons | 16 | 4 | 1.000 | 1.000 | 1.000 | 1.000 |
| breast-cancer | 286 | 9 | 0.690 | **0.707** | 0.655 | **0.672** |
| breast-cancer-wisc-prog | 198 | 33 | 0.750 | **0.800** | 0.800 | **0.825** |
| congressional-voting | 435 | 16 | 0.667 | 0.667 | 0.551 | **0.597** |
| conn-bench-sonar-mines-rocks | 208 | 60 | 0.714 | **0.786** | 0.738 | **0.833** |
| echocardiogram | 131 | 10 | 0.704 | 0.704 | 0.666 | **0.703** |
| fertility | 100 | 9 | 0.750 | **0.800** | 0.800 | 0.800 |
| haberman-survival | 306 | 3 | 0.710 | **0.774** | 0.677 | **0.709** |
| heart-hungarian | 294 | 12 | 0.831 | 0.831 | 0.779 | **0.813** |
| hepatitis | 155 | 19 | 0.645 | **0.710** | 0.709 | **0.741** |
| ionosphere | 351 | 33 | 0.887 | **0.901** | **0.929** | 0.887 |
| molec-biol-promoter | 106 | 57 | 0.818 | 0.818 | 0.727 | **0.772** |
| musk-1 | 476 | 166 | **0.958** | 0.927 | **0.947** | 0.927 |
| parkinsons | 195 | 22 | 0.974 | **1.000** | 0.923 | **1.000** |
| pittsburg-bridges-T-OR-D | 102 | 7 | 0.952 | 0.952 | 0.809 | **0.857** |
| planning | 182 | 12 | 0.541 | **0.568** | 0.649 | **0.703** |
| statlog-heart | 270 | 13 | 0.759 | **0.796** | 0.759 | **0.833** |
| trains | 10 | 29 | 1.000 | 1.000 | 1.000 | 1.000 |
| vertebral-column-2clases | 310 | 6 | 0.806 | **0.839** | 0.758 | **0.822** |
| Highest test accuracy | | | | 20/21 | | 19/21 |

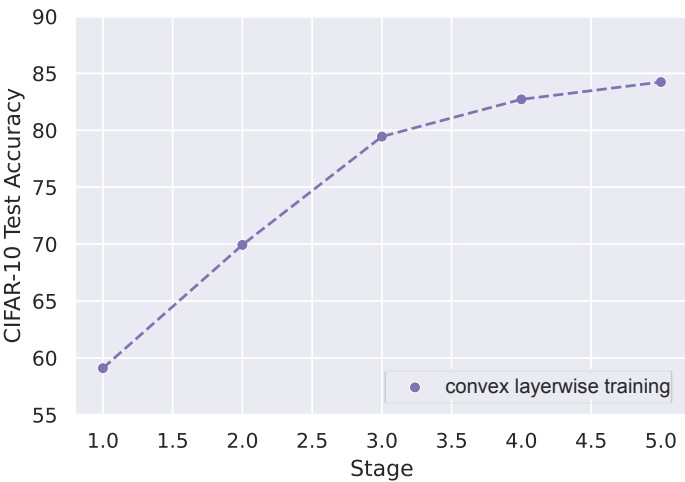

Figure 6: Test accuracy for convex layerwise training, where each stage is our three-layer convex formulation in Theorem 1. Here, we train three-layer neural networks sequentially using convex optimization to build deep networks.

**Details for the experiments in Table 2 and Figure 5:** We first note that for the experiments in Table 2, we use CVX/CVPXY (Grant & Boyd, 2014; Diamond & Boyd, 2016; Agrawal et al., 2018) to globally solve the proposed convex program in (8). For these experiments, we use a laptop with i7 processor and 16GB of RAM. In order to tune the learning rate of SGD/GD, we first perform training

with a bunch of learning different learning rates and select the one with the best performance on the validation datasets, which is $0.005$ in these experiments.

For comparatively larger scale image classification experiments in Figure 5, we utilize a cluster GPU with 50GB of memory. However since the equivalent convex program in (8) has constraint which are challenging to handle for these datasets, we propose the following unconstrained convex problem which has the same global minima with the constrained version as discussed in Gupta et al. (2021)

$$\min_{\mathbf{z},\mathbf{z}' \in \mathbb{R}^{dm_1 M P_1 P_2}} \mathcal{L}\left(\tilde{\mathbf{X}}\left(\mathbf{z}' - \mathbf{z}\right), \mathbf{y}\right) + \beta\left(\|\mathbf{z}\|_{G,1} + \|\mathbf{z}'\|_{G,1}\right) + \lambda\left(h_{\mathcal{C}}(\mathbf{z}) + h_{\mathcal{C}}(\mathbf{z}')\right) \quad (11)$$

where $\lambda > 0$ coefficient to penalize the violated constraints and $h_{\mathcal{C}}(\mathbf{z})$ is a function to sum the absolute value of all constraint violations defined as

$$h_{\mathcal{C}}(\mathbf{z}) := \mathbf{1}^T \sum_{i,j_1,l,s}\left(\left(-(2\mathbf{D}_{1ij_1} - \mathbf{I}_n)\mathbf{X}\mathbf{z}_{ij_1l}^s\right)_+\right) + \mathbf{1}^T \sum_{i,l,s}\left(-\sum_{j_1}\mathcal{I}_{ij_1l}^s(2\mathbf{D}_{2l} - \mathbf{I}_n)\mathbf{D}_{1ij_1}\mathbf{X}\mathbf{z}_{ij_1l}^s\right)_+.$$

Thus, we obtain an unconstrained version of convex optimization problem (11), where one can use commonly employed first-order gradient based optimizers, e.g., SGD and Adam, available in deep learning libraries such PyTorch and Tensorflow. For both CIFAR-10 and Fashion-MNIST, we use the same training and test splits in the original datasets. We again perform a grid search to tune the learning rate, where the best performance is achieved by the following choices

$$(\mu_{Convex}, \mu_{SGD}, \mu_{Adam}, \mu_{Adagrad}, \mu_{Adadelta}, \mu_{SGD}^{WD}) = (5e-7, 5e-3, 2e-5, 2e-3, 3e-1, 1)$$

and

$$(\mu_{Convex}, \mu_{SGD}, \mu_{Adam}, \mu_{Adagrad}, \mu_{Adadelta}, \mu_{SGD}^{WD}) = (1e-5, 2e-1, 2e-3, 1e-2, 3, 1)$$

as the learning rates for CIFAR-10 and Fashion-MNIST, respectively. We choose the momentum coefficient of the SGD optimizer as $0.9$. In addition to this, we set $P_1 P_2 = K$ and $\lambda = 1e-5$.

More importantly, we remark that these experiments are performed by using a small sampled subset of hyperplane arrangements rather than sampling all possible arrangements as detailed in Remark 3.3 of Pilanci & Ergen (2020). In particular, we first generate random weight matrices from a multivariate standard normal distribution and then solve the convex program using only the arrangements of the sampled weight matrices.

**Details for the experiments in Figure 6:** Layerwise training with shallow neural networks was proven to work remarkably well. Particularly, Belilovsky et al. (2019) shows that one can train three-layer neural networks sequentially to build deep networks that outperform end-to-end training with SOTA architectures. In Figure 6, we apply this layerwise training procedure with our convex training approach. In particular, each stage in this figure is our three-layer convex formulation in Theorem 1. Here, we use the same experimental setting in the previous section. We observe that making the network deeper by stacking convex layers resulted in significant performance improvements. Specifically, at the fifth stage, we achieved almost $85\%$ accuracy for CIFAR-10 unlike below $60\%$ accuracies in Figure 5.

## A.2 Parallel ReLU networks

The parallel networks $f_\theta(\mathbf{X})$ models a wide range of NNs in practice. As an example, standard NNs and ResNets (He et al., 2016) are special cases of this network architecture. To illustrate this, let us consider a parallel ReLU with two sub-networks and four layers, i.e., $K = 2$ and $L = 4$. If we set $\mathbf{W}_{lk} = \mathbf{W}_l \forall k \in [2], l \in [4]$ then our architecture reduces to a standard four-layer network

$$\sum_{k=1}^{2}\left(\left(\left(\mathbf{X}\mathbf{W}_{1k}\right)_+ \mathbf{W}_{2k}\right)_+ \mathbf{W}_{3k}\right)_+ \mathbf{w}_{4k} = \left(\left(\left(\mathbf{X}\mathbf{W}_1\right)_+ \mathbf{W}_2\right)_+ \mathbf{W}_3\right)_+ \mathbf{w}_4,$$

where $\mathbf{W}_1 \in \mathbb{R}^{d \times m_1}, \mathbf{W}_2 \in \mathbb{R}^{m_1 \times m_2}, \mathbf{W}_3 \in \mathbb{R}^{m_2 \times m_3}, \mathbf{w}_4 \in \mathbb{R}^{m_3}$. For ResNets, we first remark that since residual blocks are usually used after a ReLU activation function, which is positively homogeneous of degree one, in practice, each residual block takes only nonnegative entries as its inputs. Thus, we can assume $\mathbf{X} \in \mathbb{R}_+^{n \times d}$ without loss of generality. We also assume that weights

obey the following form: $\mathbf{W}_{11} = \mathbf{W}_1, \mathbf{W}_{21} = \mathbf{W}_2, \mathbf{W}_{12} = \mathbf{W}_{22} = \mathbf{I}_d, \mathbf{W}_{31} = \mathbf{W}_{32} = \mathbf{W}_3$, and $\mathbf{w}_{41} = \mathbf{w}_{42} = \mathbf{w}_4$ then

$$f_\theta(\mathbf{X}) = \sum_{k=1}^{2} \left( \left( (\mathbf{X}\mathbf{W}_{1k})_+ \mathbf{W}_{2k} \right)_+ \mathbf{W}_{3k} \right)_+ \mathbf{w}_{4k} = \left( \left( (\mathbf{X}\mathbf{W}_1)_+ \mathbf{W}_2 \right)_+ \mathbf{W}_3 \right)_+ \mathbf{w}_4 + (\mathbf{X}\mathbf{W}_3)_+ \mathbf{w}_4$$

which is a shallow ResNet as demonstrated in Figure 1 of Veit et al. (2016).

### A.3 PROOF OF LEMMA 1

Let us first define $r_{kj_{L-1}} := \sqrt{\sum_{j_1,\dots,j_{L-2}} \left( \|\mathbf{w}_{1kj_1}\|_2^2 \prod_{l=2}^{L-1} w_{lkj_{l-1}j_l}^2 \right)} > 0$. Notice that if $r_{kj_{L-1}} = 0$, this means that $k^{th}$ sub-network does not contribute the output of the parallel network in (1). Therefore, we can remove the paths with $r_{kj_{L-1}} = 0$ without loss of generality. Now, we use the following change of variable

$$\mathbf{W}'_{lk} = \mathbf{W}_{lk}, \forall l \in [L-2], \mathbf{w}'_{(L-1)kj_{L-1}} = \frac{\mathbf{W}_{(L-1)kj_{L-1}}}{r_{kj_{L-1}}}, \forall l \in [L-1], w'_{Lkj_{L-1}} = r_{kj_{L-1}} w_{Lkj_{L-1}}.$$

We now note that

$$\sum_{j_1,\dots,j_{L-2}} \left( \|\mathbf{w}'_{1kj_1}\|_2^2 \prod_{l=2}^{L-1} w'^2_{lkj_{l-1}j_l} \right) = \frac{1}{r^2_{kj_{L-1}}} \sum_{j_1,\dots,j_{L-2}} \left( \|\mathbf{w}_{1kj_1}\|_2^2 \prod_{l=2}^{L-1} w^2_{lkj_{l-1}j_l} \right) = 1.$$

Then, (2) can be restated as follows

$$p_L^* = \min_{\{\{\mathbf{W}_{lk}\}_{l=1}^{L}\}_{k=1}^{K}} \mathcal{L}\left( \sum_{k=1}^{K} \left( (\mathbf{X}\mathbf{W}_{1k})_+ \dots \mathbf{W}_{(L-1)k} \right)_+ \mathbf{w}_{Lk}, \mathbf{y} \right) + \beta \sum_{k=1}^{K} \sqrt{\sum_{j_1\dots,j_L} \left( \|\mathbf{w}_{1kj_1}\|_2^2 \prod_{l=2}^{L} w^2_{lkj_{l-1}j_l} \right)}$$

$$= \min_{\substack{\{\{\mathbf{W}_{lk}\}_{l=1}^{L-1}\}_{k=1}^{K} \\ \{\mathbf{w}'_{Lk}\}_{k=1}^{K}}} \mathcal{L}\left( \sum_{k=1}^{K} \sum_{j_{L-1}} \frac{\left( (\mathbf{X}\mathbf{W}_{1k})_+ \dots \mathbf{w}_{(L-1)kj_{L-1}} \right)_+}{r_{kj_{L-1}}} w'_{Lkj_{L-1}}, \mathbf{y} \right)$$

$$+ \beta \sum_{k=1}^{K} \sqrt{\sum_{j_{L-1}} w'^2_{Lkj_{L-1}} \frac{\sum_{j_1,\dots,j_{L-2}} \left( \|\mathbf{w}_{1kj_1}\|_2^2 \prod_{l=2}^{L-1} w^2_{lkj_{l-1}j_l} \right)}{r^2_{kj_{L-1}}}}$$

$$= \min_{\substack{\{\{\mathbf{W}_{lk}\}_{l=1}^{L-1}\}_{k=1}^{K} \\ \{\mathbf{w}'_{Lk}\}_{k=1}^{K}}} \mathcal{L}\left( \sum_{k=1}^{K} \sum_{j_{L-1}} \left( (\mathbf{X}\mathbf{W}_{1k})_+ \dots r^{-1}_{kj_{L-1}} \mathbf{w}_{(L-1)kj_{L-1}} \right)_+ w'_{Lkj_{L-1}}, \mathbf{y} \right) + \beta \sum_{k=1}^{K} \|\mathbf{w}'_{Lk}\|_2$$

$$= \min_{\substack{\{\{\mathbf{W}'_{lk}\}_{l=1}^{L}\}_{k=1}^{K} \\ (\mathbf{W}'_{1k},\dots,\mathbf{W}'_{(L-1)k}) \in \Theta_s, \forall k}} \mathcal{L}\left( \sum_{k=1}^{K} \left( (\mathbf{X}\mathbf{W}'_{1k})_+ \dots \mathbf{W}'_{(L-1)k} \right)_+ \mathbf{w}'_{Lk}, \mathbf{y} \right) + \beta \sum_{k=1}^{K} \|\mathbf{w}'_{Lk}\|_2,$$

where $\Theta_s := \left\{ (\mathbf{W}_1, \dots, \mathbf{W}_{L-1}) : \sum_{j_1,\dots,j_{L-2}} \left( \|\mathbf{w}_{1j_1}\|_2^2 \prod_{l=2}^{L-1} w^2_{lj_{l-1}j_l} \right) = 1, \forall j_{L-1} \in [m_{L-1}] \right\}$.

We also note that one can relax the equality constraint as an inequality constraint without loss of generality. This is basically due to the fact that if a constraint is not tight, i.e., strictly less than one, at the optimum then we can remove that constraint and make the corresponding output layer weight arbitrarily small via a simple scaling to make the objective value smaller. However, this would lead to a contradiction since this scaling further reduces the objective, which means that the initial set of layer weights (that yields a strict inequality in the constraints) are not optimal.

□

### A.4 PROOF OF THEOREM 1

To obtain the bidual problem of (4), we first utilize semi-infinite duality theory as follows. We first compute the dual of (6) with respect to the dual parameter $\mathbf{v}$ to get

$$p_\infty^* := \min_{\|\mathbf{w}_3\|_2 \le 1} \min_{\boldsymbol{\mu}} \mathcal{L}\left( \int_{\theta \in \Theta_s} \left( (\mathbf{X}\mathbf{W}_1)_+ \mathbf{W}_2 \right)_+ \mathbf{w}_3 d\mu(\theta), \mathbf{y} \right) + \beta \|\boldsymbol{\mu}\|_{TV}, \qquad (12)$$

where $\|\boldsymbol{\mu}\|_{TV}$ denotes the total variation norm of the signed measure $\boldsymbol{\mu}$. Notice that (12) is an infinite-dimensional neural network training problem similar to the one studied in Bach (2017). More importantly, this problem is convex since the model is linear with respect to the measure $\mu$ and the loss and regularization functions are convex (Bach, 2017). Thus, we have no duality gap, i.e., $d_3^* = p_\infty^*$. In addition to this, even though (12) is an infinite-dimensional convex optimization problem, it reduces to a problem with at most $n + 1$ neurons at the optimum due to Caratheodory's theorem (Rosset et al., 2007). Therefore, (12) can be equivalently stated as the following finite-size convex optimization problem

$$p_\infty^* = \min_{\theta \in \Theta_s} \mathcal{L}\left(\sum_{k=1}^{K^*} \left((\mathbf{X}\mathbf{W}_{1k})_+ \mathbf{W}_{2k}\right)_+ \mathbf{w}_{3k}, \mathbf{y}\right) + \beta \sum_{k=1}^{K^*} \|\mathbf{w}_{3k}\|_2$$

$$= \min_{\theta \in \Theta_s} \mathcal{L}\left(f_\theta(\mathbf{X}), \mathbf{y}\right) + \beta \sum_{k=1}^{K^*} \|\mathbf{w}_{3k}\|_2, \tag{13}$$

where $K^* \leq n + 1$. We further remark that given $K \geq K^*$, (13) and (5) are the same problems, which also proves strong duality as $p_3^* = p_\infty^* = d_3^*$. In the remainder of the proof, we show that using an alternative representation for the ReLU activation, we can achieve a finite-dimensional convex bidual formulation.

Now we restate the dual problem (6) as

$$d_3^* = \max_{\mathbf{v}} -\mathcal{L}^*(\mathbf{v}) \text{ s.t. } \max_{\theta \in \Theta_s} \left\|\mathbf{v}^T \left((\mathbf{X}\mathbf{W}_1)_+ \mathbf{W}_2\right)_+\right\|_2 \leq \beta. \tag{14}$$

We first note that using the representation in (7), the dual constraint in (14) can be written as

$$\max_{\theta \in \Theta_s} \left\|\mathbf{v}^T \left((\mathbf{X}\mathbf{W}_1)_+ \mathbf{W}_2\right)_+\right\|_2 \leq \beta \iff \max_{\theta \in \Theta_s} \sqrt{\sum_{j_2=1}^{m_2} \left(\mathbf{v}^T \left((\mathbf{X}\mathbf{W}_1)_+ \mathbf{w}_{2j_2}\right)_+\right)^2} \leq \beta$$

$$\iff \max_{\theta \in \Theta_s} \sqrt{m_2 \left(\mathbf{v}^T \left((\mathbf{X}\mathbf{W}_1)_+ \mathbf{w}_2\right)_+\right)^2} \leq \beta$$

$$\iff \max_{\mathcal{I}_{j_1} \in \{\pm 1\}} \max_{\theta \in \Theta_s} \sqrt{m_2 \left(\mathbf{v}^T \left(\sum_{j_1=1}^{m_1} \mathcal{I}_{j_1} \left(\mathbf{X}\mathbf{w}_{1j_1} | w_{2j_1}|\right)_+\right)_+\right)^2} \leq \beta$$

$$\iff \max_{\substack{i \in [P_1] \\ l \in [P_2]}} \max_{\mathcal{I}_{j_1} \in \{\pm 1\}} \max_{\{\mathbf{w}_{j_1}\}_{j_1} \in \mathcal{C}_{il}} \sqrt{m_2} \left|\mathbf{v}^T \mathbf{D}_{2l} \sum_{j_1=1}^{m_1} \mathcal{I}_{j_1} \mathbf{D}_{1ij_1} \mathbf{X}\mathbf{w}_{j_1}\right| \leq \beta \tag{15}$$

where $\Theta_s = \{(\mathbf{W}_1, \mathbf{W}_2) : \sum_{j_1=1}^{m_1} \|\mathbf{w}_{1j_1}\|_2^2 w_{2j_1 j_2}^2 \leq 1, \forall j_2 \in [m_2]\}$, we apply a variable change as $\mathbf{w}_{j_1} = |w_{2j_1}| \mathbf{w}_{1j_1}$ and define the set $\mathcal{C}_{il}$ as

$$\mathcal{C}_{il} := \{\{\mathbf{w}_{j_1}\}_{j_1} : (2\mathbf{D}_{2l} - \mathbf{I}_n) \sum_{j_1=1}^{m_1} \mathcal{I}_{j_1} \mathbf{D}_{1ij_1} \mathbf{X}\mathbf{w}_{j_1} \geq 0, (2\mathbf{D}_{1ij_1} - \mathbf{I}_n)\mathbf{X}\mathbf{w}_{j_1} \geq 0, \forall j_1 \in [m_1], \sum_{j_1=1}^{m_1} \|\mathbf{w}_{j_1}\|_2^2 \leq 1\}.$$

We also note that $P_1$ and $P_2$ denote the number of possible hyperplane arrangement for the first and second ReLU layer (see Appendix A.10 for details).

Then we have

$$\max_{\theta \in \Theta_s} \left\|\mathbf{v}^T \left((\mathbf{X}\mathbf{W}_1)_+ \mathbf{W}_2\right)_+\right\|_2 \leq \beta \iff \max_{\substack{i \in [P_1] \\ l \in [P_2] \\ \mathcal{I}_{j_1} \in \{\pm 1\}}} \max_{\{\mathbf{w}_{j_1}\}_{j_1} \in \mathcal{C}_{il}} \sqrt{m_2} \left|\mathbf{v}^T \mathbf{D}_{2l} \sum_{j_1=1}^{m_1} \mathcal{I}_{j_1} \mathbf{D}_{1ij_1} \mathbf{X}\mathbf{w}_{j_1}\right| \leq \beta,$$

$$\iff \begin{aligned} &\max_{\{\mathbf{w}_{ij_1 l}^s\}_{j_1} \in \mathcal{C}_{il}^s} \sqrt{m_2} \mathbf{v}^T \mathbf{D}_{2l} \sum_{j_1=1}^{m_1} \mathcal{I}_{ij_1 l}^s \mathbf{D}_{1ij_1} \mathbf{X}\mathbf{w}_{ij_1 l}^s \leq \beta, \\ &\phantom{xxxxxxxxxxxxxxxxxxxxxxxxxxxxxxx}, \forall i \in [P_1], \forall l \in [P_2], \forall s \in [M], \\ &\max_{\{\mathbf{w}_{ij_1 l}^{s'}\}_{j_1} \in \mathcal{C}_{il}^s} -\sqrt{m_2} \mathbf{v}^T \mathbf{D}_{2l} \sum_{j_1=1}^{m_1} \mathcal{I}_{ij_1 l}^{s'} \mathbf{D}_{1ij_1} \mathbf{X}\mathbf{w}_{ij_1 l}^{s'} \leq \beta, \end{aligned}$$

where we use $M := |\{\pm 1\}^{m_1}| = 2^{m_1}$ to enumerate all possible sign patterns $\{\mathcal{I}_{j_1}\}_{j_1=1}^{m_1}$ of the size $m_1$. Using the equivalent representation above, we rewrite the dual (14) as

$$\max_{\mathbf{v}} -\mathcal{L}^*(\mathbf{v}) \text{ s.t. } \max_{\{\mathbf{w}_{ij_1l}^s\}_{j_1} \in \mathcal{C}_{il}^s} \sqrt{m_2} \mathbf{v}^T \mathbf{D}_{2l} \sum_{j_1=1}^{m_1} \mathcal{I}_{ij_1l}^s \mathbf{D}_{1ij_1} \mathbf{X} \mathbf{w}_{ij_1l}^s \le \beta, \ \forall i, l, s \tag{16}$$

$$\max_{\{\mathbf{w}_{ij_1l}^{s'}\}_{j_1} \in \mathcal{C}_{il}^s} -\sqrt{m_2} \mathbf{v}^T \mathbf{D}_{2l} \sum_{j_1=1}^{m_1} \mathcal{I}_{ij_1l}^{s'} \mathbf{D}_{1ij_1} \mathbf{X} \mathbf{w}_{ij_1l}^{s'} \le \beta, \ \forall i, l, s. \tag{17}$$

Since the problem above is convex and satisfies the Slater's condition when all the parameters are set to zero, we have strong duality (Boyd & Vandenberghe, 2004), and thus we can state (16) as

$$\min_{\substack{\gamma_{il}^s \ge 0 \\ \gamma_{il}^{s'} \ge 0}} \max_{\mathbf{v}} \min_{\substack{\{\mathbf{w}_{ij_1l}^s\}_{j_1} \in \mathcal{C}_{il}^s \\ \{\mathbf{w}_{ij_1l}^{s'}\}_{j_1} \in \mathcal{C}_{il}^s}} -\mathcal{L}(\mathbf{v})^* + \sum_{s=1}^M \sum_{i=1}^{P_1} \sum_{l=1}^{P_2} \gamma_{il}^s \left( \beta - \sqrt{m_2} \mathbf{v}^T \mathbf{D}_{2l} \sum_{j_1=1}^{m_1} \mathcal{I}_{ij_1l}^s \mathbf{D}_{1ij_1} \mathbf{X} \mathbf{w}_{ij_1l}^s \right) \tag{18}$$

$$+ \sum_{s=1}^M \sum_{i=1}^{P_1} \sum_{l=1}^{P_2} \gamma_{il}^{s'} \left( \beta + \sqrt{m_2} \mathbf{v}^T \mathbf{D}_{2l} \sum_{j_1=1}^{m_1} \mathcal{I}_{ij_1l}^{s'} \mathbf{D}_{1ij_1} \mathbf{X} \mathbf{w}_{ij_1l}^{s'} \right). \tag{19}$$

Due to Sion's minimax theorem (Sion, 1958), we can change the order the inner minimization and maximization to obtain closed-form solutions for the maximization over the variable $\mathbf{v}$. This yields the following problem

$$\min_{\substack{\gamma_{il}^{su} \ge 0 \\ }} \min_{\substack{\{\mathbf{w}_{ij_1l}^s\}_{j_1} \in \mathcal{C}_{il}^s \\ \{\mathbf{w}_{ij_1l}^{s'}\}_{j_1} \in \mathcal{C}_{il}^s}} \mathcal{L} \left( \sum_{s=1}^M \sum_{l=1}^{P_2} \sum_{i=1}^{P_1} \sum_{j_1=1}^{m_1} \sqrt{m_2} \mathbf{D}_{2l} \mathbf{D}_{1ij_1} \mathbf{X} (\mathcal{I}_{ij_1l}^s \gamma_{il}^s \mathbf{w}_{ij_1l}^s - \mathcal{I}_{ij_1l}^{s'} \gamma_{il}^s \mathbf{w}_{ij_1l}^{s'}), \mathbf{y} \right)$$

$$+ \beta \sum_{s=1}^M \sum_{i=1}^{P_1} \sum_{l=1}^{P_2} (\gamma_{il}^s + \gamma_{il}^{s'}). \tag{20}$$

Finally, we introduce a set of variable changes changes as $\mathbf{z}_{ij_1l}^s = \sqrt{m_2} \gamma_{il}^s \mathcal{I}_{ij_1l}^s \mathbf{w}_{ij_1l}^s$ and $\mathbf{z}_{ij_1l}^{s'} = \sqrt{m_2} \gamma_{il}^{s'} \mathcal{I}_{ij_1l}^{s'} \mathbf{w}_{ij_1l}^{s'}$ such that (20) can be cast as the following convex problem

$$\min_{\substack{\{\mathbf{z}_{ij_1l}^s\}_{j_1} \in \mathcal{C}_{il}^{s'} \\ \{\mathbf{z}_{ij_1l}^{s'}\}_{j_1} \in \mathcal{C}_{il}^{s'}}} \mathcal{L} \left( \sum_{s=1}^M \sum_{l=1}^{P_2} \sum_{i=1}^{P_1} \sum_{j_1=1}^{m_1} \mathbf{D}_{2l} \mathbf{D}_{1ij_1} \mathbf{X} (\mathbf{z}_{ij_1l}^s - \mathbf{z}_{ij_1l}^{s'}), \mathbf{y} \right)$$

$$+ \frac{\beta}{\sqrt{m_2}} \sum_{s=1}^M \sum_{i=1}^{P_1} \sum_{l=1}^{P_2} \left( \sqrt{\sum_{j_1=1}^{m_1} \|\mathbf{z}_{ij_1l}^s\|_2^2} + \sqrt{\sum_{j_1=1}^{m_1} \|\mathbf{z}_{ij_1l}^{s'}\|_2^2} \right), \tag{21}$$

where the constraint set $\mathcal{C}_{il}^s$ are defined as

$$\mathcal{C}_{il}^{s'} := \left\{ \{\mathbf{z}_{j_1}\}_{j_1} : (2\mathbf{D}_{2l} - \mathbf{I}_n) \sum_{j_1=1}^{m_1} \mathbf{D}_{1ij_1} \mathbf{X} \mathbf{z}_{j_1} \ge 0, \ (2\mathbf{D}_{1ij_1} - \mathbf{I}_n) \mathbf{X} \mathcal{I}_{ij_1l}^s \mathbf{z}_{j_1} \ge 0, \forall j_1 \in [m_1] \right\}.$$

Notice that (21) is a constrained convex optimization problem with $2dm_1 M P_1 P_2$ variables and $2n(m_1 + 1) M P_1 P_2$ constraints in the set $\mathcal{C}_{il}^s$.

$\square$

## A.5 PROOF OF PROPOSITION 1

In this section, we prove that once the convex program in (21) is globally optimized to obtain a set of optimal solutions $\{\mathbf{z}_{ij_1l}^{s*}, \mathbf{z}_{ij_1l}^{s'*}\}_{i,j_1,l,s}$, one can recover an optimal solution to the non-convex

training problem (4) via a simple closed-form mapping as detailed below

$$\mathbf{W}_{1k}^* = \begin{cases} \frac{1}{m_2} \begin{bmatrix} \mathcal{I}_{i1l}^s \mathbf{z}_{i1l}^{s^*} & \mathcal{I}_{i2l}^s \mathbf{z}_{i2l}^{s^*} & \cdots & \mathcal{I}_{im_1l}^s \mathbf{z}_{im_1l}^{s^*} \end{bmatrix} & \text{if } 1 \le k \le MP_1P_2 \\ \frac{1}{m_2} \begin{bmatrix} \mathcal{I}_{i1l}^{s'} \mathbf{z}_{i1l}^{s'^*} & \mathcal{I}_{i2l}^{s'} \mathbf{z}_{i2l}^{s'^*} & \cdots & \mathcal{I}_{im_1l}^{s'} \mathbf{z}_{im_1l}^{s'^*} \end{bmatrix} & \text{if } MP_1P_2 + 1 \le k \le 2MP_1P_2 \end{cases}$$

$$\mathbf{W}_{2k}^* = \begin{cases} \begin{bmatrix} \mathcal{I}_{i1l}^s & \mathcal{I}_{i1l}^s & \cdots & \mathcal{I}_{11l}^s \\ \vdots & \vdots & \cdots & \vdots \\ \mathcal{I}_{im_1l}^s & \mathcal{I}_{im_1l}^s & \cdots & \mathcal{I}_{1m_1l}^s \end{bmatrix} & \text{if } 1 \le k \le MP_1P_2 \\ \\ \begin{bmatrix} \mathcal{I}_{i1l}^{s'} & \mathcal{I}_{i1l}^{s'} & \cdots & \mathcal{I}_{11l}^{s'} \\ \vdots & \vdots & \cdots & \vdots \\ \mathcal{I}_{im_1l}^{s'} & \mathcal{I}_{im_1l}^{s'} & \cdots & \mathcal{I}_{1m_1l}^{s'} \end{bmatrix} & \text{if } MP_1P_2 + 1 \le k \le 2MP_1P_2 \end{cases}$$

$$\mathbf{w}_{3k}^* = \begin{cases} \begin{bmatrix} 1 & 1 & \cdots & 1 \end{bmatrix}^T & \text{if } 1 \le k \le MP_1P_2 \\ \begin{bmatrix} -1 & -1 & \cdots & -1 \end{bmatrix}^T & \text{if } MP_1P_2 + 1 \le k \le 2MP_1P_2 \end{cases},$$

where

$$(s,l,i) = \begin{cases} \left( \left\lfloor \frac{k-1}{P_1P_2} \right\rfloor + 1, \left\lfloor \frac{k-1-(s-1)P_1P_2}{P_1} \right\rfloor + 1, k - (s-1)P_1P_2 - (l-1)P_1 \right) & \text{if } 1 \le k \le MP_1P_2 \\ \left( \left\lfloor \frac{k'-1}{P_1P_2} \right\rfloor + 1, \left\lfloor \frac{k'-1-(s-1)P_1P_2}{P_1} \right\rfloor + 1, k' - (s-1)P_1P_2 - (l-1)P_1 \right) & \text{else} \end{cases}$$

with $k' = k - MP_1P_2$. Hence, we achieve an optimal solution to the original non-convex training problem (4) as $\{\mathbf{W}_{1k}^*, \mathbf{W}_{2k}^*, \mathbf{w}_{3k}^*\}_{k=1}^{2MP_1P_2}$, where $\mathbf{W}_{1k}^* \in \mathbb{R}^{d \times m_1}$, $\mathbf{W}_{2k}^* \in \mathbb{R}^{m_1 \times m_2}$, and $\mathbf{w}_{3k}^* \in \mathbb{R}^{m_2}$ respectively. Next, we confirm that the proposed set of layer weights are indeed optimal by plugging them back to both the convex and non-convex objectives.

We first verify that both the optimal convex and non-convex layer weights give the same network output as follows (8), i.e.,

$$\sum_{k=1}^{2MP_1P_2} \left( (\mathbf{X}\mathbf{W}_{1k}^*)_+ \mathbf{W}_{2k}^* \right)_+ \mathbf{w}_{3k}^* = \sum_{s=1}^{M} \sum_{l=1}^{P_2} \sum_{i=1}^{P_1} \sum_{j=1}^{m_1} \mathbf{D}_{2l} \mathbf{D}_{1ij} \mathbf{X} \left( \mathbf{z}_{ij_1l}^{s^*} - \mathbf{z}_{ij_1l}^{s'^*} \right).$$

Now, we show that the proposed set of weight matrices for the non-convex problem achieves the same regularization cost with (21), i.e.,

$$\sum_{k=1}^{2MP_1P_2} \sqrt{\sum_{j_2=1}^{m_2} \sum_{j_1=1}^{m_1} \|\mathbf{w}_{1kj_1}^*\|_2^2 w_{2kj_1j_2}^{*2} w_{3kj_2}^{*2}} = \frac{1}{\sqrt{m_2}} \sum_{s=1}^{M} \sum_{i=1}^{P_1} \sum_{l=1}^{P_2} \left( \sqrt{\sum_{j_1=1}^{m_1} \|\mathbf{z}_{ij_1l}^{s^*}\|_2} + \sqrt{\sum_{j=1}^{m_1} \|\mathbf{z}_{ij_1l}^{s'^*}\|_2} \right).$$

Since $\{\mathbf{W}_{1k}^*, \mathbf{W}_{2k}^*, \mathbf{w}_{3k}^*\}_{k=1}^{2MP_1P_2}$ yields the same network output and regularization cost with the optimal parameters of the convex program in (21), we conclude that the proposed set parameters for the non-convex problem also achieves the optimal objective value $p_3^*$, i.e.,

$$p_3^* = \mathcal{L} \left( \sum_{k=1}^{2MP_1P_2} \left( (\mathbf{X}\mathbf{W}_{1k}^*)_+ \mathbf{W}_{2k}^* \right)_+ \mathbf{w}_{3k}^*, \mathbf{y} \right) + \beta \sum_{k=1}^{2MP_1P_2} \sqrt{\sum_{j_2=1}^{m_2} \sum_{j_1=1}^{m_1} \|\mathbf{w}_{kj_1}^*\|_2^2 w_{2kj_1j_2}^{*2} w_{3kj_2}^{*2}}.$$

$\square$

## A.6 PROOF OF THEOREM 2

We start with defining the optimal parameters for the original and rank-$k$ approximation of the rescaled problem in (5) as

$$\{(\mathbf{W}_{1k}^*, \mathbf{W}_{2k}^*, \mathbf{w}_{3k}^*)\}_{k=1}^K := \underset{\theta \in \Theta_s}{\arg\min} \, \mathcal{L} \left( \sum_{k=1}^K \left( (\mathbf{X}\mathbf{W}_{1k})_+ \mathbf{W}_{2k} \right)_+ \mathbf{w}_{3k}, \mathbf{y} \right) + \beta \sum_{k=1}^K \|\mathbf{w}_{3k}\|_2$$

$$\{(\hat{\mathbf{W}}_{1k}, \hat{\mathbf{W}}_{2k}, \hat{\mathbf{w}}_{3k})\}_{k=1}^K := \underset{\theta \in \Theta_s}{\arg\min} \, \mathcal{L} \left( \sum_{k=1}^K \left( (\hat{\mathbf{X}}_r \mathbf{W}_{1k})_+ \mathbf{W}_{2k} \right)_+ \mathbf{w}_{3k}, \mathbf{y} \right) + \beta \sum_{k=1}^K \|\mathbf{w}_{3k}\|_2$$

(22)

and the objective value achieved by the parameters trained using $\hat{\mathbf{X}}_r$ as

$$p_r := \mathcal{L}\left(\sum_{k=1}^{K}\left(\left(\mathbf{X}\hat{\mathbf{W}}_{1k}\right)_+\hat{\mathbf{W}}_{2k}\right)_+\hat{\mathbf{w}}_{3k}, \mathbf{y}\right) + \beta\sum_{k=1}^{K}\|\hat{\mathbf{w}}_{3k}\|_2.$$

Then, we have

$$p_3^* = \mathcal{L}\left(\sum_{k=1}^{K}\left(\left(\mathbf{X}\mathbf{W}_{1k}^*\right)_+\mathbf{W}_{2k}^*\right)_+\mathbf{w}_{3k}^*, \mathbf{y}\right) + \beta\sum_{k=1}^{K}\|\mathbf{w}_{3k}^*\|_2$$

$$\overset{(i)}{\leq} \mathcal{L}\left(\sum_{k=1}^{K}\left(\left(\mathbf{X}\hat{\mathbf{W}}_{1k}\right)_+\hat{\mathbf{W}}_{2k}\right)_+\hat{\mathbf{w}}_{3k}, \mathbf{y}\right) + \beta\sum_{k=1}^{K}\|\hat{\mathbf{w}}_{3k}\|_2 = p_r$$

$$\overset{(ii)}{\leq} \mathcal{L}\left(\sum_{k=1}^{K}\left(\left(\hat{\mathbf{X}}_r\hat{\mathbf{W}}_{1k}\right)_+\hat{\mathbf{W}}_{2k}\right)_+\hat{\mathbf{w}}_{3k}, \mathbf{y}\right) + (\beta + \sqrt{m_1 m_2}R\sigma_{r+1})\sum_{k=1}^{K}\|\hat{\mathbf{w}}_{3k}\|_2$$

$$\leq \left(\mathcal{L}\left(\sum_{k=1}^{K}\left(\left(\hat{\mathbf{X}}_r\hat{\mathbf{W}}_{1k}\right)_+\hat{\mathbf{W}}_{2k}\right)_+\hat{\mathbf{w}}_{3k}, \mathbf{y}\right) + \beta\sum_{k=1}^{K}\|\hat{\mathbf{w}}_{3k}\|_2\right)\left(1 + \frac{\sqrt{m_1 m_2}R\sigma_{r+1}}{\beta}\right)$$

$$\overset{(iii)}{\leq} \left(\mathcal{L}\left(\sum_{k=1}^{K}\left(\left(\hat{\mathbf{X}}_r\mathbf{W}_{1k}^*\right)_+\mathbf{W}_{2k}^*\right)_+\mathbf{w}_{3k}^*, \mathbf{y}\right) + \beta\sum_{k=1}^{K}\|\mathbf{w}_{3k}^*\|_2\right)\left(1 + \frac{\sqrt{m_1 m_2}R\sigma_{r+1}}{\beta}\right)$$

$$\overset{(iv)}{\leq} \left(\mathcal{L}\left(\sum_{k=1}^{K}\left(\mathbf{X}\mathbf{W}_{1k}^*\right)_+\mathbf{W}_{2k}^*\right)_+\mathbf{w}_{3k}^*, \mathbf{y}\right) + \beta\sum_{k=1}^{K}\|\mathbf{w}_{3k}^*\|_2\right)\left(1 + \frac{\sqrt{m_1 m_2}R\sigma_{r+1}}{\beta}\right)^2$$

$$= p_3^*\left(1 + \frac{\sqrt{m_1 m_2}R\sigma_{r+1}}{\beta}\right)^2,$$

where $(i)$ and $(iii)$ follow from the optimality definitions of the original and approximated problems in (22). In addition, $(ii)$ and $(iv)$ follow from the relations below

$$\mathcal{L}\left(\sum_{k=1}^{K}\left(\left(\mathbf{X}\hat{\mathbf{W}}_{1k}\right)_+\hat{\mathbf{W}}_{2k}\right)_+\hat{\mathbf{w}}_{3k}, \mathbf{y}\right)$$

$$= \mathcal{L}\left(\sum_{k=1}^{K}\sum_{j_2=1}^{m_2}\left[\left(\left(\mathbf{X}\hat{\mathbf{W}}_{1k}\right)_+\hat{\mathbf{w}}_{2kj_2}\right)_+\hat{w}_{3kj_2} - \left(\left(\hat{\mathbf{X}}_r\hat{\mathbf{W}}_{1k}\right)_+\hat{\mathbf{w}}_{2kj_2}\right)_+\hat{w}_{3kj_2} + \left(\left(\hat{\mathbf{X}}_r\hat{\mathbf{W}}_{1k}\right)_+\hat{\mathbf{w}}_{2kj_2}\right)_+\hat{w}_{3kj_2}\right], \mathbf{y}\right)$$

$$\overset{(1)}{\leq} \mathcal{L}\left(\sum_{k=1}^{K}\sum_{j_2=1}^{m_2}\left[\left(\left(\mathbf{X}\hat{\mathbf{W}}_{1k}\right)_+\hat{\mathbf{w}}_{2kj_2}\right)_+\hat{w}_{3kj_2} - \left(\left(\hat{\mathbf{X}}_r\hat{\mathbf{W}}_{1k}\right)_+\hat{\mathbf{w}}_{2kj_2}\right)_+\hat{w}_{3kj_2}\right], \mathbf{y}\right) + \mathcal{L}\left(\sum_{k=1}^{K}\sum_{j_2=1}^{m_2}\left(\left(\hat{\mathbf{X}}_r\hat{\mathbf{W}}_{1k}\right)_+\hat{\mathbf{w}}_{2kj_2}\right)_+\hat{w}_{3kj_2}, \mathbf{y}\right)$$

$$\overset{(2)}{\leq} R\left\|\sum_{k=1}^{K}\sum_{j_2=1}^{m_2}\left[\left(\left(\mathbf{X}\hat{\mathbf{W}}_{1k}\right)_+\hat{\mathbf{w}}_{2kj_2}\right)_+\hat{w}_{3kj_2} - \left(\left(\hat{\mathbf{X}}_r\hat{\mathbf{W}}_{1k}\right)_+\hat{\mathbf{w}}_{2kj_2}\right)_+\hat{w}_{3kj_2}\right]\right\|_2 + \mathcal{L}\left(\sum_{k=1}^{K}\sum_{j_2=1}^{m_2}\left(\left(\hat{\mathbf{X}}_r\hat{\mathbf{W}}_{1k}\right)_+\hat{\mathbf{w}}_{2kj_2}\right)_+\hat{w}_{3kj_2}, \mathbf{y}\right)$$

$$= R\left\|\sum_{k=1}^{K}\sum_{j_2=1}^{m_2}\left(\left(\left(\mathbf{X}\hat{\mathbf{W}}_{1k}\right)_+\hat{\mathbf{w}}_{2kj_2}\right)_+ - \left(\left(\hat{\mathbf{X}}_r\hat{\mathbf{W}}_{1k}\right)_+\hat{\mathbf{w}}_{2kj_2}\right)_+\right)\hat{w}_{3kj_2}\right\|_2 + \mathcal{L}\left(\sum_{k=1}^{K}\sum_{j_2=1}^{m_2}\left(\left(\hat{\mathbf{X}}_r\hat{\mathbf{W}}_{1k}\right)_+\hat{\mathbf{w}}_{2kj_2}\right)_+\hat{w}_{3kj_2}, \mathbf{y}\right)$$

$$\overset{(3)}{\leq} R\sum_{k=1}^{K}\sum_{j_2=1}^{m_2}\left\|\left(\left(\mathbf{X}\hat{\mathbf{W}}_{1k}\right)_+\hat{\mathbf{w}}_{2kj_2}\right)_+ - \left(\left(\hat{\mathbf{X}}_r\hat{\mathbf{W}}_{1k}\right)_+\hat{\mathbf{w}}_{2kj_2}\right)_+\right\|_2 |\hat{w}_{3kj_2}| + \mathcal{L}\left(\sum_{k=1}^{K}\sum_{j_2=1}^{m_2}\left(\left(\hat{\mathbf{X}}_r\hat{\mathbf{W}}_{1k}\right)_+\hat{\mathbf{w}}_{2kj_2}\right)_+\hat{w}_{3kj_2}, \mathbf{y}\right)$$

$$\leq R\max_{k\in[K], j_2\in[m_2]}\left\|\left(\left(\mathbf{X}\hat{\mathbf{W}}_{1k}\right)_+\hat{\mathbf{w}}_{2kj_2}\right)_+ - \left(\left(\hat{\mathbf{X}}_r\hat{\mathbf{W}}_{1k}\right)_+\hat{\mathbf{w}}_{2kj_2}\right)_+\right\|_2 \sum_{k=1}^{K}\|\hat{\mathbf{w}}_{3k}\|_1 + \mathcal{L}\left(\sum_{k=1}^{K}\sum_{j_2=1}^{m_2}\left(\left(\hat{\mathbf{X}}_r\hat{\mathbf{W}}_{1k}\right)_+\hat{\mathbf{w}}_{2kj_2}\right)_+\hat{w}_{3kj_2}, \mathbf{y}\right)$$

$$\overset{(4)}{\leq} R\max_{k\in[K]}\left\|\mathbf{X} - \hat{\mathbf{X}}_r\right\|_2 \sum_{j_1=1}^{m_1}\|\hat{\mathbf{w}}_{1kj_1}\|_2 |\hat{w}_{2kj_1j_2}| \sum_{k=1}^{K}\|\hat{\mathbf{w}}_{3k}\|_1 + \mathcal{L}\left(\sum_{k=1}^{K}\sum_{j_2=1}^{m_2}\left(\left(\hat{\mathbf{X}}_r\hat{\mathbf{W}}_{1k}\right)_+\hat{\mathbf{w}}_{2kj_2}\right)_+\hat{w}_{3kj_2}, \mathbf{y}\right)$$

$$\overset{(5)}{\leq} \sqrt{m_1}R\max_{k\in[K]}\left\|\mathbf{X} - \hat{\mathbf{X}}_r\right\|_2 \sqrt{\sum_{j_1=1}^{m_1}\|\hat{\mathbf{w}}_{1kj_1}\|_2^2 |\hat{w}_{2kj_1j_2}|^2} \sum_{k=1}^{K}\|\hat{\mathbf{w}}_{3k}\|_1 + \mathcal{L}\left(\sum_{k=1}^{K}\sum_{j_2=1}^{m_2}\left(\left(\hat{\mathbf{X}}_r\hat{\mathbf{W}}_{1k}\right)_+\hat{\mathbf{w}}_{2kj_2}\right)_+\hat{w}_{3kj_2}, \mathbf{y}\right)$$

$$\overset{(6)}{=} \sqrt{m_1} R\sigma_{r+1} \sum_{k=1}^{K} \|\hat{\mathbf{w}}_{3k}\|_1 + \mathcal{L}\left( \sum_{k=1}^{K} \sum_{j_2=1}^{m_2} \left( \left( \hat{\mathbf{X}}_r \hat{\mathbf{W}}_{1k} \right)_+ \hat{\mathbf{w}}_{2kj_2} \right)_+ \hat{w}_{3kj_2}, \mathbf{y} \right)$$

$$\leq \sqrt{m_1 m_2} R\sigma_{r+1} \sum_{k=1}^{K} \|\hat{\mathbf{w}}_{3k}\|_2 + \mathcal{L}\left( \sum_{k=1}^{K} \sum_{j_2=1}^{m_2} \left( \left( \hat{\mathbf{X}}_r \hat{\mathbf{W}}_{1k} \right)_+ \hat{\mathbf{w}}_{2kj_2} \right)_+ \hat{w}_{3kj_2}, \mathbf{y} \right),$$

where we use the convexity and $R$-Lipschtiz property of the loss function, convexity of $\ell_2$-norm, 1-Lipschitz property of the ReLU activation, $\|\mathbf{x}\|_1 \leq \sqrt{m}\|\mathbf{x}\|_2$ for $\mathbf{x} \in \mathbb{R}^m$, and $\max_k \sum_{j_1=1}^{m_1} \|\hat{\mathbf{w}}_{1kj_1}\|_2^2 \hat{w}_{2kj_1j_2}^2 \leq 1$ from the rescaling in Lemma 1 for $(1), (2), (3), (4), (5),$ and $(6)$ respectively. $\qquad\square$

### A.7 PROOF FOR THE DUAL PROBLEM IN (3)

In order to prove the dual problem, we directly utilize Fenchel duality (Boyd & Vandenberghe, 2004). Let us first rewrite the primal regularized training problem after the application of the rescaling in Lemma 1 as follows

$$p_L^* = \min_{\hat{\mathbf{y}} \in \mathbb{R}^n, \theta \in \Theta_s} \mathcal{L}\left( \hat{\mathbf{y}}, \mathbf{y} \right) + \beta \sum_{k=1}^{K} \|\mathbf{w}_{Lk}\|_2 \text{ s.t. } \hat{\mathbf{y}} = \sum_{k=1}^{K} \left( (\mathbf{X}\mathbf{W}_{1k})_+ \ldots \mathbf{W}_{(L-1)k} \right)_+ \mathbf{w}_{Lk}. \quad (23)$$

The corresponding Lagrangian can be computed by incorporating the constraints into objective via a dual veriable as follows

$$L(\mathbf{v}, \hat{\mathbf{y}}, \mathbf{w}_{Lk}) = \mathcal{L}\left( \hat{\mathbf{y}}, \mathbf{y} \right) - \mathbf{v}^T \hat{\mathbf{y}} + \mathbf{v}^T \sum_{k=1}^{K} \left( (\mathbf{X}\mathbf{W}_{1k})_+ \ldots \mathbf{W}_{(L-1)k} \right)_+ \mathbf{w}_{Lk} + \beta \sum_{k=1}^{K} \|\mathbf{w}_{Lk}\|_2.$$

We then define the following dual function

$$g(\mathbf{v}) = \min_{\hat{\mathbf{y}}, \mathbf{w}_{Lk}} L(\mathbf{v}, \hat{\mathbf{y}}, \mathbf{w}_{Lk})$$

$$= \min_{\hat{\mathbf{y}}, \mathbf{w}_{Lk}} \mathcal{L}\left( \hat{\mathbf{y}}, \mathbf{y} \right) - \mathbf{v}^T \hat{\mathbf{y}} + \mathbf{v}^T \sum_{k=1}^{K} \left( (\mathbf{X}\mathbf{W}_{1k})_+ \ldots \mathbf{W}_{(L-1)k} \right)_+ \mathbf{w}_{Lk} + \beta \sum_{k=1}^{K} \|\mathbf{w}_{Lk}\|_2$$

$$= -\mathcal{L}^*(\mathbf{v}) \text{ s.t. } \left\| \mathbf{v}^T \left( (\mathbf{X}\mathbf{W}_{1k})_+ \ldots \mathbf{W}_{(L-1)k} \right)_+ \right\|_2 \leq \beta, \forall k \in [K],$$

where $\mathcal{L}^*$ is the Fenchel conjugate function defined as (Boyd & Vandenberghe, 2004)

$$\mathcal{L}^*(\mathbf{v}) := \max_{\mathbf{z}} \mathbf{z}^T \mathbf{v} - \mathcal{L}\left( \mathbf{z}, \mathbf{y} \right).$$

Hence, we write the dual problem of (23) as

$$p_L^* = \min_{\theta \in \Theta_s} \max_{\mathbf{v}} g(\mathbf{v}) = \min_{\theta \in \Theta_s} \max_{\mathbf{v}} -\mathcal{L}^*(\mathbf{v}) \text{ s.t. } \left\| \mathbf{v}^T \left( (\mathbf{X}\mathbf{W}_{1k})_+ \ldots \mathbf{W}_{(L-1)k} \right)_+ \right\|_2 \leq \beta, \forall k \in [K].$$

However, since the hidden layer weights are the variables of the outer minimization, we cannot directly characterize the optimal hidden layer weight in the form above. Thus, as the last step of the derivation, we change the order of the minimization over $\theta$ and the maximization over $\mathbf{v}$ to obtain the following lower bound

$$p_L^* \geq d_L^* = \max_{\mathbf{v}} \min_{\theta \in \Theta_s} -\mathcal{L}^*(\mathbf{v}) \text{ s.t. } \left\| \mathbf{v}^T \left( (\mathbf{X}\mathbf{W}_{1k})_+ \ldots \mathbf{W}_{(L-1)k} \right)_+ \right\|_2 \leq \beta, \forall k \in [K]$$

$$= \max_{\mathbf{v}} -\mathcal{L}^*(\mathbf{v}) \text{ s.t. } \max_{\theta \in \Theta_s} \left\| \mathbf{v}^T \left( (\mathbf{X}\mathbf{W}_1)_+ \ldots \mathbf{W}_{(L-1)} \right)_+ \right\|_2 \leq \beta.$$

$$\square$$

### A.8 HYPERPLANE ARRANGEMENTS

Here, we review the notion of hyperplane arrangements detailed in Pilanci & Ergen (2020).

We first define the set of all hyperplane arrangements for the data matrix $\mathbf{X}$ as

$$\mathcal{H} := \bigcup \{\{\text{sign}(\mathbf{X}\mathbf{w})\} : \mathbf{w} \in \mathbb{R}^d\},$$

where $|\mathcal{H}| \leq 2^n$. The set $\mathcal{H}$ all possible $\{+1, -1\}$ labelings of the data samples $\{\mathbf{x}_i\}_{i=1}^n$ via a linear classifier $\mathbf{w} \in \mathbb{R}^d$. We now define a new set to denote the indices with positive signs for each element in the set $\mathcal{H}$ as $\mathcal{S} := \{\{\cup_{h_i=1}\{i\}\} : \mathbf{h} \in \mathcal{H}\}$. With this definition, we note that given an element $S \in \mathcal{S}$, one can introduce a diagonal matrix $\mathbf{D}(S) \in \mathbb{R}^{n \times n}$ defined as

$$\mathbf{D}(S)_{ii} := \begin{cases} 1 & \text{if } i \in S \\ 0 & \text{otherwise} \end{cases}.$$

$\mathbf{D}(S)$ can be also viewed as a diagonal matrix of indicators, where each diagonal entry is one if the the corresponding sample labeled as $+1$ by the linear classifier $\mathbf{w}$, zero otherwise. Therefore, the output of ReLU activation can be equivalently written as $(\mathbf{Xw})_+ = \mathbf{D}(S)\mathbf{Xw}$ provided that $\mathbf{D}(S)\mathbf{Xw} \geq 0$ and $(\mathbf{I}_n - \mathbf{D}(S))\mathbf{Xw} \leq 0$ are satisfied. One can define more compactly these two constraints as $(2\mathbf{D}(S) - \mathbf{I}_n)\mathbf{Xw} \geq 0$. We now denote the cardinality of $\mathcal{S}$ as $P$, and obtain the following upperbound

$$P \leq 2 \sum_{k=0}^{r-1} \binom{n-1}{k} \leq 2r \left(\frac{e(n-1)}{r}\right)^r$$

where $r := \text{rank}(\mathbf{X}) \leq \min(n, d)$ (Ojha, 2000; Stanley et al., 2004; Winder, 1966; Cover, 1965).

## A.9 LOW RANK MODEL IN THEOREM 2

In Section 3.1, we propose an $\epsilon$-approximate training approach that has polynomial-time complexity even when the data matrix is full rank. Here, you can select the rank $r$ by plugging in the desired approximation error and network structure in equation 10. We show that the approximation error proved in Theorem 2 can be arbitrarily small for practically relevant problems. As an example, consider a parallel architecture training problem with $\ell_2$ loss function, then the upperbound becomes $(1 + \frac{\sqrt{m_1 m_2}\sigma_{r+1}}{\beta})^2$, which can be arbitrarily close to one due to presence of noise component (with small $\sigma_{r+1}$) in most datasets in practice (see Figure 4 for an empirical verification). This observation is also valid for several benchmark datasets, including MNIST, CIFAR-10, and CIFAR-100, which exhibit exponentially decaying singular values (see Figure 7) and therefore effectively has a low rank structure. In addition, singular values can be computed to set the target rank and the value of the regularization coefficient to obtain any desired approximation ratio using Theorem 2.

## A.10 PROOF OF PROPOSITION 2 AND COROLLARY 1

We first review the multi-layer hyperplane arrangements concept introduced in Section 2.1 of Ergen & Pilanci (2021c). Based on this concept, we then calculate the training complexity to globally solve the convex program in Theorem 1.

If we denote the number of hyperplane arrangements for the first ReLU layer as $P_1$, then from Appendix A.8 we know that

$$P_1 \leq 2r \left(\frac{e(n-1)}{r}\right)^r \approx \mathcal{O}(n^r). \tag{24}$$

In order to obtain a bound for the number of hyperplane arrangements in the second ReLU layer we first note that preactivations of the second ReLU layer, i.e., $(\mathbf{XW}_1)_+ \mathbf{w}_2$ can be equivalently represented as a matrix-vector product form by using the effective data matrix $\bar{\mathbf{X}} := [\mathcal{I}_1 \mathbf{D}_{11}\mathbf{X} \quad \mathcal{I}_2\mathbf{D}_{12}\mathbf{X} \quad \ldots \quad \mathcal{I}_{m_1}\mathbf{D}_{1m_1}\mathbf{X}]$ due to the equivalent representation in (7). Therefore, given $\text{rank}(\bar{\mathbf{X}}) = r_2 \leq m_1 r$

$$\bar{P}_2 \leq 2 \sum_{k=0}^{r_2-1} \binom{n-1}{k} \leq 2r_2 \left(\frac{e(n-1)}{r_2}\right)^{r_2} \leq 2m_1 r \left(\frac{e(n-1)}{m_1 r}\right)^{m_1 r}.$$

However, notice that $\bar{\mathbf{X}}$ is not a fixed data matrix since we can choose each diagonal $\mathbf{D}_{1j_1}$ among a set $\{\mathbf{D}_i\}_{i=1}^{P_1}$ of size $P_1$ due to (24) and the sign pattern $\mathcal{I}_{j_1}$ among the set $\{+1, -1\}$ of size 2. Thus, in the worst-case, we have the following upper-bound

$$P_2 \leq \bar{P}_2(2P_1)^{m_1} \leq \frac{2^{2m_1+1}(e(n-1))^{2m_1 r}}{m_1^{m_1 r-1} r^{2m_1 r-m_1-1}} \approx \mathcal{O}(n^{m_1 r}). \tag{25}$$

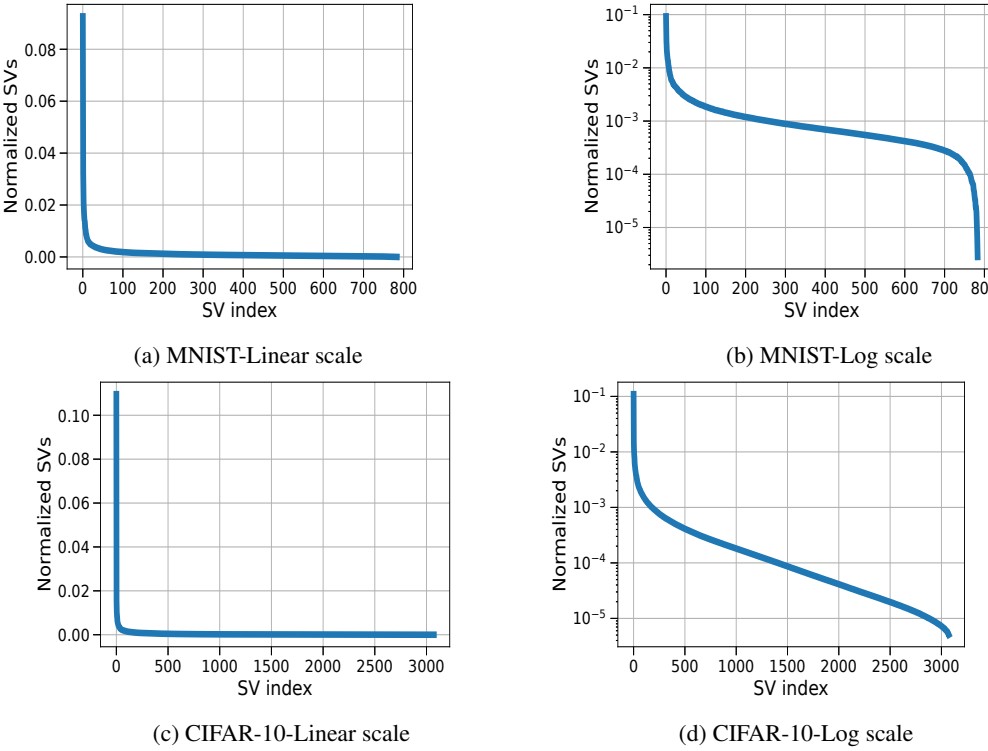

(a) MNIST-Linear scale                (b) MNIST-Log scale

(c) CIFAR-10-Linear scale         (d) CIFAR-10-Log scale

Figure 7: The values of normalized singular values of the data matrix for the MNIST and CIFAR-10 datasets. As illustrated in both figures, the singular values follow an exponentially decaying trends indicating an effective low rank structure.

Notice that given fixed scalars $m_1$ and $r$, both $P_1$ and $P_2$ are polynomial terms with respect to the number of data samples $n$ and the feature dimension $d$.

**Remark 3.** *Notice that Convolutional Neural Networks (CNNs) operate on the patch matrices $\{\mathbf{X}_b\}_{b=1}^B$ instead of the full data matrix $\mathbf{X}$, where $\mathbf{X}_b \in \mathbb{R}^{n \times h}$ and $h$ denotes the filter size. Hence, even when the data matrix is full rank, i.e., $r = \min\{n, d\}$, the number of hyperplane arrangements $P_1$ is upperbounded as $P_1 \leq \mathcal{O}(n^{r_c})$, where $r_c := \max_b rank(\mathbf{X}_b) \leq h \ll \min\{n, d\}$ (see Ergen & Pilanci (2021a) for details). For instance, let us consider a CNN with $m_1 = 512$ filters of size $3 \times 3$, then $r_c \leq 9$ independent of data dimension $n, d$. As a consequence, weight sharing structure in CNNs dramatically limits the number of possible hyperplane arrangements. This also explains efficiency and remarkable generalization performance of CNNs in practice.*

**Training complexity analysis:** Here, we calculate the computational complexity to globally solve the convex program in (8). Note that (8) is a convex optimization problem with $2dm_1MP_1P_2$ variables and $2n(m_1 + 1)MP_1P_2$ constraints. Therefore, due to the upperbounds in (24) and (25) of Appendix A.8, the convex program (8) can be globally optimized by a standard interior-point solver with the computational complexity $\mathcal{O}(d^3m_1^3 2^{3(m_1+1)} n^{3(m_1+1)r})$, which is a polynomial-time complexity in terms of $n, d$.

The analysis in this section can be recursively extended to arbitrarily deep parallel networks. First notice that if we apply the same approach to obtain an upperbound on $P_3$, then due to the multiplicative pattern in (25), we obtain $P_3 \leq P_3'(2P_2)^{m_2} \leq \mathcal{O}(n^{m_2m_1r})$. In a similar manner, the number of hyperplane arrangements in the $l^{th}$ layer is upperbounded as $P_l \leq \mathcal{O}(n^{r \prod_{j=1}^{l-1} m_j})$, which is polynomial in both $n$ and $d$ for fixed data rank $r$ and fixed layer widths $\{m_j\}_{j=1}^{l-1}$.

$\square$

## A.11 Extension to vector outputs

In this section, we extend the analysis to parallel networks with multiple outputs where the label matrix is defined as $\mathbf{Y} \in \mathbb{R}^{n \times C}$ provided that there exist $C$ classes/outputs. Then the primal non-convex training problem is as follows

$$p_v^* := \min_{\theta \in \Theta} \mathcal{L}\left(\sum_{k=1}^{K} f_{\theta,k}(\mathbf{X}), \mathbf{Y}\right) + \beta \sum_{k=1}^{K} \sqrt{\sum_{j_1, j_2, \ldots, j_L} \left(\|\mathbf{w}_{1kj_1}\|_2^2 \prod_{l=2}^{L-1} w_{lkj_{l-1}j_l}^2 \|\mathbf{w}_{Lkj_{L-1}j_L}^2\|^2\right)}.$$

Applying Lemma 1 and each step in the proof of Theorem 1 yield

$$d_v^* := \max_{\mathbf{V}} \min_{\theta \in \Theta_s} -\mathcal{L}^*(\mathbf{V}) \text{ s.t. } \left\|\mathbf{V}^T\left((\mathbf{X}\mathbf{W}_{1k})_+ \ldots \mathbf{W}_{(L-1)k}\right)_+\right\|_F \leq \beta, \ \forall k \in [K],$$

where the corresponding Fenchel conjugate function is

$$\mathcal{L}^*(\mathbf{V}) := \max_{\mathbf{Z}} \operatorname{trace}\left(\mathbf{Z}^T \mathbf{V}\right) - \mathcal{L}(\mathbf{Z}, \mathbf{Y}).$$

Notice that above we have a dual matrix $\mathbf{V}$ instead of the dual vector in the scalar output case. More importantly, here, we have $\ell_2$ norm in the dual constraint unlike the scalar output case with absolute value. Therefore, the vector-output case is slightly more challenging than the scalar output case and yields a different regularization function in the equivalent convex program.

The rest of the derivations directly follows the steps in Section A.4 and Sahiner et al. (2021b).

