# OpenReview forum: "Path Regularization: A Convexity and Sparsity Inducing Regularization for Parallel ReLU Networks"
_ICLR.cc/2023/Conference — Submitted to ICLR 2023_

### Official Review · Reviewer_QEB5 · 2022-10-24

**Confidence:** 3
**Correctness:** 3
**Technical Novelty And Significance:** 3
**Empirical Novelty And Significance:** Not applicable
**Recommendation:** 6

**Clarity, Quality, Novelty And Reproducibility:**

The paper is mostly well written, although there are some flaw in the paper. The results presented in this paper is novel.

**Strength And Weaknesses:**

The paper proposed a new method to solve three-layer parallel neural network training problem with path regularization, and experiments show that this method outperform gradient descent-style algorithms in speed and accuracy.
There are some flaws in the paper. The path regularization given in Table 2 in different from that in equation 4 by a square root. Is something missing here?
The proof of Lemma 1 is not clear to me. $w$ and $w’$ in the third to the last equation in page 16 have 3 subscripts each, while those in the second to the last equation in this page have 4. Besides, the  second to the last equation in this page does not make sense to me.
Some equations in the appendix go beyond the page boundary. Eg. the last set of equations in page 16.
Section A.1 mention that results in Figure 5 are computed by converting the constraint optimization problem into an unconstrained one so SGD can be used. I would suggest to move this part to the main paper and thoroughly discuss how to implement this, eg. how to select $\lambda$.
In Figure 5, I suggest the author to add the results of weight decay using Adam, etc., which reflects the most common empirical methods. The accuracy on cifar-10 dataset is too low to make much sense. I would suggest to replace it with MNIST, for example.
Also, I am curious whether this method can be extended to neural networks beyond 3 layers.

**Summary Of The Paper:**

This paper presented a method to convert the non-convex optimization problem of parallel neural network trained with path regularization into a convex optimization problem, and proposed a low rank approximation to the model such that the model can be trained in polynomial time. Experiments show that the proposed method can achieve higher accuracy than gradient descent-style algorithms in the same amount of time.

**Summary Of The Review:**

The paper proposed a new method to train a neural network . The author provided good theoretical support to this method.
This method yields higher accuracy and speed in a toy dataset experiments and small image classification tasks. However, there are some flaw in the paper which should be fixed.

---

### Official Review · Reviewer_ctH6 · 2022-10-30

**Confidence:** 2
**Correctness:** 2
**Technical Novelty And Significance:** 3
**Empirical Novelty And Significance:** 3
**Recommendation:** 6

**Clarity, Quality, Novelty And Reproducibility:**

The mathematical presentation is quite poor plus several ambiguities in the text.

I have some doubts about the validity of the results. See comments below
I am willing to change my rating if the authors clarify the issues

**Strength And Weaknesses:**

The main result, provably polynomial time training for neural nets, is quite strong and novel.

The major weakness is that the paper is very hard to read, and the authors should benefit from simplifying the notation and improving the clarity.

**Summary Of The Paper:**

This paper proposes an "equivalent" convex formulation to a family of regularized neural network (with ReLU activation) optimization problems. This has very nice consequences such as interpreting neural network as sparsification machine in overparameterized regime or polynomial time trainability. The authors demonstrate numerically the advantages of considering such formulation.

**Summary Of The Review:**

1) The path regularization (which is not explicitly defined in the paper) might be ill defined. Based on Lemma 1, I guess the authors refer to the path $\beta \mapsto \hat \theta(\beta)$ where $\hat \theta(\beta)$ is an element of the argmin of the regularized solution. However, the non-uniqueness of the solution implies that the notion of path regularization is ill defined here (a single beta mapsto different solutions).

2) Ambiguity with the notion of “exact solution”.  Standard convex solvers do not provide exact solutions but only approximate to a prescribed tolerance. Can the authors clarify this point specially with respect to proposition 2. More specifically, what does it mean to “solve” the optimization problem globally?  Note that even for vanilla Lasso problem, exact solutions cannot be obtained without exponential complexity in general (explicit worst case example exists.)

3) In proposition 1, provide explicitly the mapping between the convex solution and “the” non-convex one. Since the latter is non-unique, the authors might want to clarify the meaning here. Also, since the exact solution cannot be obtained, how the approximation error affects the claim in the proposition 1.

4) The alternative representation for ReLU in section 3 might be incorrect. Can the authors explain the first equality?
The product is $XW_1$ first. I don't understand how the index $j_1$ travels inside the positive part and affects only $W_1$.


Minor:
- Define precisely $\Theta$
- The constants $P_1, P_2$ are not defined?
- Why the bidual objective coincide with the primal? Is it because of strong duality? Can the authors detail that part a bit.
- The sentence "Based on the analysis above, exponential complexity is unavoidable for deep networks when the data
matrix is full rank" is confusing since the authors just argue that the complexity is polynomial.
- After Eq. 6, "Therefore, we first show that strong duality holds in this case, i.e.,", it should be an equality
- The first sentence after Eq 6 is very confusing too.
- In figure 3, what does the points $(1, 0), (2,2), (3,3)$ represent? The message of this figure is quite unclear.
- Define $h_C$ in Eq. 11
- After Eq. 14, the index is missing on $XW$ ? Why does the sum disappear in the second equivalence?
- More comments on the generalization to more than 3 Layers

---

### Official Review · Reviewer_mUtW · 2022-10-31

**Confidence:** 4
**Clarity, Quality, Novelty And Reproducibility:** See above, paragraphe "weaknesses"
**Correctness:** 2
**Technical Novelty And Significance:** 3
**Empirical Novelty And Significance:** 2
**Recommendation:** 3

**Strength And Weaknesses:**

Strengths:
The training of deep neural networks is a major topic of interest in the current literature. Understanding how neural networks generalize so well despite the non convex cost function and abundance of bad local minimizers is a critical question.
Highlighting hidden convexity in such training is a promising direction.

Weaknesses:
- The "polynomial" claim is based on a trick where the design matrix is approximated by a low rank one through a truncated SVD (Eq 10).
Thus **only an approximation is solved**. The original problem remains of exponential complexity as one would expect, and as is mentioned in Table 1. The paper's claim is thus exaggerated in my opinion: the **original** problem is not solved in polynomial time, as there is no indication that $r \ll \min (n, d)$ in the general case.
Nevertheless the paper explicitly writes several times sentences such as "We prove that training the path regularized parallel ReLU networks (1) is equivalent to a convex optimization problem that can be solved in polynomial-time by standard convex solvers". That is not correct.

The paper cannot be accepted in its current form due to these claims. Removing them, and saying "The problem can be approximately solved in polynomial time" would fix the issue.

- The exponential dependency on the product over the layers of the numbers of hidden neurons also prevents the method from being used practically. I believe it is still of interesting theoretical nature, but the authors may consider acknowledging this fact in the paper.

**Summary Of The Paper:**

The paper studies the training of deep parallel ReLU networks with L2 regularization/weight decay.
As has been done for 2 layers Relu networks by Pilanci and Ergen (2020), it derives an equivalent convex reformulation of the original optimization problem. This reformulation involves a group sparsity penalty, that generalizes the results of Pilanci and Ergen (2020).

The technique is the same as in the original paper: use positive homogeneity of the ReLU to derive an equivalent constrained formulation (Lemma 1), then derive the dual of this formulation and show that strong duality holds, so that the solution of the derived dual can be mapped back to a global solution of the primal. The derivation is based on diagonal binary matrices $D_i$ that correspond to activation patterns of the ReLU layers.
The novelty compared to the original paper is the grouping effect over parallel networks, and the extended number of layers.


**Summary Of The Review:**

Approach marginally improving on known results using same duality techniques. Claims of polynomial time solving are overstated and experiments make it clear that the proposed method is still more of theoretical interest than of practical one.

---

### Official Review · Reviewer_n2WM · 2022-11-06

**Confidence:** 4
**Correctness:** 3
**Technical Novelty And Significance:** 2
**Empirical Novelty And Significance:** 2
**Recommendation:** 6

**Clarity, Quality, Novelty And Reproducibility:**

The work was generally clear of good quality, although I was confused at times (see comments above).

I am unsure what is new/different in this paper over Wang et al.'21 since they also prove results on zero duality gap for parallel networks (in fact, with any number of layers).


**Strength And Weaknesses:**


### Weaknesses / places I was confused

* I am confused about the duality claimed in (3) and proved in Appendix A.7. It seems that the authors only prove weak duality ($p_L^* \geq d_L^*$), but then in the subsequent paragraphs they claim that they prove strong duality $p_L^* = d_L^*$ as guaranteed in Lemma 1. However, the Lemma 1 seems to me unrelated to the arguments in Appendix A.7? In Appendix A.4 it is shown there is no duality gap, but only for the case of L = 3 layers. I would also appreciate clarification with what is new here compared to the paper Wang et al. '21, which is cited at the end and proves there is no duality gap in the same setting, but for any L. (Also, Corollary 1 of this paper states that the analysis extends to any number of layers, but no proof is given.)

* The authors state that ResNets are special cases of this parallel network architecture. However, I am unconvinced by this because it requires tying weights (see Appendix A.2) and fixing weights to equal the identity in the parallel network. Doesn't this mean that the optimization problem for finding the minimum-cost ResNet is completely different from the optimization problem for finding the minimum-cost parallel network architecture?

* I found Theorem 1 to be difficult to read. Several new variables have been introduced right before stating this theorem. Specifically "we first enumerate all possible signs and diagonal matrices for the ReLU layers and denote them as $ I_{j_1}, D_{1ij_{1}}$,
and $D_{2l}$ respectively, where $j_1 \in [m_1], i \in [P_1], l \in [P_2]$". This notation is not very clear to me, and also it took some work to understand that the authors must assume that the data lies in a rank $r$ subspace of the $d$-dimensional input space, where effectively $r = O(1)$ for the algorithm to run in polynomial time.

* Since the data must lies in a $r = O(1)$-dimensional subspace of $\mathbb{R}^d$ for the exact algorithm to run in polynomial time, it seems like a stretch to advertise in the abstract that the algorithm runs in polynomial-time in feature dimension. This is true, but if the rank $r \ll d$, it seems that one can preprocess the data and project it onto the lower-dimensional space and now treat $d$ as $d = O(1)$.

* The sentence "Based on the analysis above, exponential complexity is unavoidable for deep networks when the data matrix is full rank, i.e., $rank(X) = min(n, d)$" is not justified. The previous analysis was an upper bound on running times, and it does not say anything about the (unlikely) possibility that there is a better algorithm that runs in time $\mathrm{poly}(n,d)$ time for full-rank data.

* Section 3.2 claims "Here, we provide a complete explanation for the representational power of three-layer networks by comparing with the two-layer results in Pilanci & Ergen (2020)". However, (1) it is already known in the literature that two-layer networks are universal representers, so Section 3.2 must be about the cost of representing with 2 vs. 3 layers. And (2) it is already known in the literature that there are functions that take exponentially more neurons to represent with 2 layers than with 3 layers. In short, I am unsure what is contributed by Section 3.2.

* Finally, why do the authors take a path regularization instead of an L2 regularization? It would be nice to have an intuitive explanation of
why the former yields a better convex problem than the latter.

### Strengths

* An approximate algorithm is given with a runtime that depends as $n^{\kappa}$, where $\kappa$ effective rank of the data. This addresses one of the weaknesses above. The idea here is nice: take a low-rank approximation of the data, and run the exact convex solver on it.

* The authors provide experiments on small datasets and small networks, showing that learning with the convex program can yield better/faster solutions than training with SGD or Adam. I found Table 3 in the appendix particularly impressive. However, I would like to know -- if you take a very wide, overparametrized network, and train it with SGD/Adam, will it still be outperformed by the convex program?

### Typos
* Footnote 2: "all proofs are provided in the supplementary file", should say in appendix A.3
* "with three-layer layers"
* why is $w_{3k} \in \mathbb{R}^K$ ? I think this is a typo
* "therefore we show that strong duality holds in this case, i.e., $p_3^* \geq d_3^*$ as detailed in Appendix A.4". Should be $p_3^* = d_3^*$

**Summary Of The Paper:**

This paper studies convex formulations of the problem of training parallel ReLU networks.

These are models of the form \sum_{i=1}^K f_i(x; \theta^i), where each f_i is a ReLU network.

The problem gives algorithms for exactly/approximately learning these networks with path regularization.

The result is by a convex formulation of the networks, extending results of Pilanci and Ergen'20 beyond 2 layers.

**Summary Of The Review:**

I would be willing to accept this paper if the authors could clarify my confusions and address the weaknesses I have listed above.

---

### Decision · Program_Chairs · 2023-01-20

**Decision:**

Reject

**Justification For Why Not Higher Score:**

It could've been getting accepted if there is a slot.  There are some concrete deltas over existing work.

**Justification For Why Not Lower Score:**

N/A

**Metareview: Summary, Strengths And Weaknesses:**

The paper considers a parallel ReLU network with path regularization and shows that it's equivalent to a convex program by showing that the duality gap is 0. The specific form of the convex program is worked out, which indicates that it is inducing a group-sparsity regularization term.  In addition, a layer-wise training algorithm was proposed that leverages the convex reformulation as well as an approximation scheme. The main claim is that one gets a polynomial-time approximation to a problem known to be hard. Experiments show computational benefits over standard SGD on a toyscale dataset.

Reviewers cite several weaknesses, including:
- The algorithm is overall exponential time (in the feature dimension or the intrinsic rank), so the theoretical advantage over the gradient-based method is unclear.
- The convex reformulation is only valid for up to three-layer NNs and the authors admit some challenges in generalizing to deeper networks.
- Some overclaims on a number of matters, e.g., "polynomial time" algorithms;  mixing the representation-power of the parallel DNN (can represent ResNet,  standard DNN, etc) with a particular regularized solution.
- Several reviewers also find the papers hard to read and had a lot of confusion initially.




**Summary Of Ac-Reviewer Meeting:**

The authors did a good job addressing some of the issues in the revised version of the paper. Notably, the empirical results on CIFAR10 got better after the authors tried to stack more layers. However, the reviewers still find that the novelty is incremental over existing work. Ultimately, no reviewer was able to champion the paper.

I think the paper can benefit from a round of revision that removes all confusing statements / overclaiming then it will be a strong submission to ICML. If some progress can be made on the case of L>3 using ideas from Wang et al. (2021), then the paper would be a very nice contribution.